# Enhanced growth rate of atmospheric particles from sulphuric acid

Dominik Stolzenburg[1,2], Mario Simon[3], Ananth Ranjithkumar[4], Andreas Kürten[3], Katrianne Lehtipalo[2,5], Hamish Gordon[4], Sebastian Ehrhart[6], Henning Finkenzeller[7], Lukas Pichelstorfer[2], Tuomo Nieminen[2], Xu-Cheng He[2], Sophia Brilke[1], Mao Xiao[8], António Amorim[9], Rima Baalbaki[2], Andrea Baccarini[8], Lisa Beck[2], Steffen Bräkling[10], Lucía Caudillo Murillo[3], Dexian Chen[11], Biwu Chu[2], Lubna Dada[2], António Dias[9], Josef Dommen[8], Jonathan Duplissy[2], Imad El Haddad[8], , Lukas Fischer[12], Loic Gonzalez Carracedo[1], Martin Heinritzi[3], Changhyuk Kim[13,14], Theodore K. Koenig[7], Weimeng Kong[13], Houssni Lamkaddam[8], Chuan Ping Lee[8], Markus Leiminger[12,15], Zijun Li[16], Vladimir Makhmutov[17], Hanna E. Manninen[18], Guillaume Marie[3], Ruby Marten[8], Tatjana Müller[3], Wei Nie[19], Eva Partoll[12], Tuukka Petäjä[2], Joschka Pfeifer[18], Maxim Philippov[17], Matti P. Rissanen[2,20], Birte Rörup[2], Siegfried Schobesberger[16], Simone Schuchmann[18], Jiali Shen[2], Mikko Sipilä[2], Gerhard Steiner[12], Yuri Stozhkov[17], Christian Tauber[1], Yee Jun Tham[2], António Tomé[21], Miguel Vazquez-Pufleau[1], Andrea C. Wagner[3,7], Mingyi Wang[11], Yonghong Wang[2], Stefan K. Weber[18], Daniela Wimmer[1,2], Peter J. Wlasits[1], Yusheng Wu[2], Qing Ye[11], Marcel Zauner-Wieczorek[3], Urs Baltensperger[8], Kenneth S. Carslaw[4], Joachim Curtius[3], Neil M. Donahue[11], Richard C. Flagan[13], Armin Hansel[12,15], Markku Kulmala[2], Jos Lelieveld[6], Rainer Volkamer[7], Jasper Kirkby[3,18] and Paul M. Winkler[1]

[1]Faculty of Physics, University of Vienna, 1090 Vienna, Austria.
[2]Institute for Atmospheric and Earth System Research/Physics, University of Helsinki, 00014 Helsinki, Finland.
[3]Institute for Atmospheric and Environmental Sciences, Goethe University Frankfurt, 60438 Frankfurt am Main, Germany.
[4]School of Earth and Environment, University of Leeds, Leeds LS2 9JT, United Kingdom.
[5]Finnish Meteorological Institute, 00560 Helsinki, Finland.
[6]Atmospheric Chemistry Department, Max-Planck-Institute for Chemistry, 55128 Mainz, Germany
[7]Department of Chemistry and Cooperative Institute for Research in Environmental Sciences, University of Colorado Boulder, 80309 Boulder CO, USA.
[8]Laboratory of Atmospheric Chemistry, Paul Scherrer Institute, 5232 Villigen, Switzerland.
[9]Center for Astrophysics and Gravitation, Faculty of Sciences of the University of Lisbon, 1749-016 Lisbon, Portugal.
[10]Tofwerk AG, 3600 Thun, Switzerland.
[11]Center for Atmospheric Particle Studies, Carnegie Mellon University, 15217 Pittsburgh PA, USA.
[12]Institute for Ion Physics and Applied Physics, University of Innsbruck, 6020 Innsbruck, Austria.
[13]Division of Chemistry and Chemical Engineering, California Institute of Technology, 91125 Pasadena CA, USA.
[14]Department of Environmental Engineering, Pusan National University, Busan 46241, Republic of Korea.
[15]Ionicon Analytik GmbH, 6020 Innsbruck, Austria.
[16]Department of Applied Physics, University of Eastern Finland, 70211 Kuopio, Finland.
[17]P.N. Lebedev Physical Institute of the Russian Academy of Sciences, 119991 Moscow, Russia.
[18]CERN, the European Organization for Nuclear Research, 1211 Geneva, Switzerland.
[19]Joint International Research Laboratory of Atmospheric and Earth System Sciences, School of Atmospheric Sciences, Nanjing University, 210023 Nanjing, China.
[20]Aerosol Physics Laboratory, Tampere University, 33101 Tampere, Finland.
[21]Institute Infante Dom Luíz, University of Beira Interior, 6200-001 Covilhã, Portugal.

*Correspondence to*: Paul M. Winkler (paul.winkler@univie.ac.at)

**Abstract.** In the present-day atmosphere, sulphuric acid is the most important vapour for aerosol particle formation and initial growth. However, the growth rates of nanoparticles (<10 nm) from sulphuric acid remain poorly measured. Therefore, the effect of stabilizing bases, the contribution of ions and the impact of attractive forces on molecular collisions are under debate. Here we present precise growth-rate measurements of uncharged sulphuric acid particles from 1.8-10 nm, performed under atmospheric conditions in the CERN CLOUD chamber. Our results show that the evaporation of sulphuric acid particles above 2 nm is negligible and growth proceeds kinetically even at low ammonia concentrations. The experimental growth rates exceed the hard-sphere kinetic limit for condensation of sulphuric acid. We demonstrate that this results from van-der-Waals forces between the vapour molecules and particles and disentangle it from charge-dipole interactions. The magnitude of the enhancement depends on the assumed particle hydration and collision kinetics, but is increasingly important at smaller sizes resulting in a steep rise of observed growth rates with decreasing size. Including the experimental results in a global model, we find the enhanced growth rate of sulphuric acid particles increases predicted particle number concentrations in the upper free troposphere by more than 50%.

## 1 Introduction

Sulphuric acid ($H_2SO_4$) is the major atmospheric trace compound responsible for nucleation of aerosol particles in the present-day atmosphere (Dunne et al., 2016). Sulphuric acid participates in new particle formation (NPF) in the upper troposphere (Brock et al., 1995; Weber et al., 1999; Weigel et al., 2011), stratosphere (Deshler, 2008), polar regions (Jokinen et al., 2018), urban or anthropogenic influenced environments (Yao et al., 2018) and when a complex mixture of different condensable vapours is present (Lehtipalo et al., 2018). Especially in the initial growth of small atmospheric molecular clusters, sulphuric acid is likely of crucial importance (Kulmala et al., 2013). The newly formed particles need to grow rapidly in order to avoid scavenging by larger, pre-existing aerosols and, thereby, contribute to the global cloud condensation nuclei (CCN) budget (Pierce and Adams, 2007). The dynamics in this cluster size-range of a few nm therefore determines the climatic significance of atmospheric NPF, which is the major source of CCN (Gordon et al., 2017) and can also affect urban air quality (Guo et al., 2014).

The main pathway of cluster and particle growth is condensation of low volatility vapours, like sulphuric acid or oxidized organics (Stolzenburg et al., 2018). Nanoparticle growth rates depend both on the evaporation rates of the condensing vapours and on the molecular collision frequencies. Uncertainty about the expected behaviour at the collision ("kinetic") limit influences the interpretation of experimental data. One focus has been on evaporation rates from small particles and potential growth-rate enhancement from coagulation. It has been shown in earlier laboratory measurements that bases like ammonia can have a stabilizing effect for growth below 2 nm (Lehtipalo et al., 2016). If amines, which are stronger bases than ammonia, are added, nucleation itself can proceed at the kinetic limit, i.e. evaporation rates from the monomer onwards are zero (Jen et al., 2014; Kürten et al., 2014; Olenius et al., 2013). In this case, cluster coagulation also plays an important role in the growth

process due to the strong clustering behaviour of sulphuric acid and amines (Kontkanen et al., 2016; Lehtipalo et al., 2016; Li and McMurry, 2018). However, in the presence of ammonia, the evaporation rates and the magnitude of cluster coagulation remain unmeasured, although ammonia is much more important than amines globally due to its longer atmospheric lifetime. A second focus is on the collisional rate coefficients themselves, which may be enhanced by either charge-dipole interactions (Nadykto and Yu, 2003) or van-der-Waals forces (Chan and Mozurkewich, 2001). In spite of the importance of these coefficients there are only few direct measurements of the charge effect on growth (Lehtipalo et al., 2016; Svensmark et al., 2017). Even if the charge-dipole interactions are stronger, an enhancement due to van-der-Waals forces might be more important at typical atmospheric ionization levels. Several atmospheric studies have demonstrated that sulphuric acid uptake proceeds close at a collision-limited rate (Bzdek et al., 2013; Kuang et al., 2010), but they could neither provide a measurement of a collision enhancement nor considered in detail hydration effects (Verheggen and Mozurkewich, 2002). Both might be significant in the free molecular regime below 5 nm, where growth measurements are also affected by larger uncertainties (Kangasluoma and Kontkanen, 2017). Here, we address the questions of evaporation and collision enhancement in sulphuric acid driven growth with precision measurements (Stolzenburg et al., 2017) at the CERN (European Organization for Nuclear Research) CLOUD experiment (Duplissy et al., 2016).

## 2 Methods

### 2.1 Experimental approach

The CERN CLOUD chamber is a 26.1 m$^3$ stainless steel aerosol chamber, which can be kept at a constant temperature within 0.1 K precision. It offers the possibility to study new particle formation under different ionization levels. Two high voltage electrode grids inside the chamber can efficiently clear ions and charged particles from the chamber within seconds, ensuring neutral conditions. When there is no electric field in the chamber galactic cosmic rays lead to an ion production rate of ~2-4 ion pairs cm$^{-3}$ s$^{-1}$. Ion concentrations can also be elevated to upper tropospheric conditions by illumination of the chamber with a pion beam from the CERN proton-synchrotron. The dry air supply for the chamber is provided by boil-off oxygen and boil-off nitrogen mixed at the atmospheric ratio of 79:21. This ensures extremely low contaminant levels, especially from organics and sulphuric acid. This was verified by a PTR3 proton-transfer-reaction time of flight mass spectrometer (Breitenlechner et al., 2017) and a nitrate chemical ionization-atmospheric pressure interface-time of flight mass spectrometer (nitrate CI-APi-ToF) (Jokinen et al., 2012). The absence of any contamination from amines was confirmed by measurements with a water cluster-CI-APi-ToF (Pfeifer et al., 2019), which did not register dimethylamine mixing ratios above the detection limit of 0.1 pptv.

We performed measurements of particle growth from sulphuric acid and ammonia at either +20 °C or +5°C with the relative humidity kept constant at either 38% or 60%. SO$_2$ (5 ppb), O$_3$ (~120 ppb) and ammonia (varied between 3 and 1000 pptv) were injected into the chamber. The experiments were initiated by homogeneous illumination of the chamber at constant O$_3$ and SO$_2$ levels. The UV light of four Hamamatsu UV lamps guided into the chamber with fibre optics induced the photo-

dissociation of $O_3$ and production of OH· radicals. Thereby, $SO_2$ is oxidized, leading to the formation of sulphuric acid (varied between $10^7$ and $10^9$ cm$^{-3}$). A typical experiment is shown in Fig. S1 in the Supplement. Sulphuric acid monomer concentrations were measured with the nitrate CI-API-TOF. Calibration of the instrument's response to sulphuric acid (Kürten et al., 2012) was performed before and after the measurement campaign and yielded comparable results. Compared to previous

studies, also the measurement of gas-phase $NH_3$ significantly improved due to the deployment of the calibrated water cluster CI-APi-ToF. The protonated water cluster reagent ions selectively ionize ammonia and amines at ambient pressure reaching a detection limit of approximately 0.5 pptv for ammonia.

Particle growth was monitored using a differential mobility analyser-train (DMA-train) (Stolzenburg et al., 2017) for the main size-range of 1.8-8 nm. We also include measurements from a Caltech nano-radial DMA (Brunelli et al., 2009) with a custom-

build DEG-counter for sizes between 4-8 nm and a TSI Model 3936 nano-SMPS for sizes larger than 5 nm when investigating the size-dependency of the growth. For the growth of the charged fraction, we use a neutral cluster and air ion spectrometer (NAIS) (Manninen et al., 2009). All four instruments use electrical mobility classification and measured mobility diameters are corrected to mass diameters (Larriba et al., 2011) for the calculation of collision kinetics. Compared to the scanning particle-size-magnifier (see e.g. Lehtipalo et al., 2014), which was used in Lehtipalo et al. (2016), these instruments using

direct mobility analysis have less systematic uncertainty on the actual size classification. The size-ranges of both studies are also not directly comparable. We show the measurements in the lower size-interval of the DMA-train (1.8-3.2 nm mobility diameter) together with the earlier results (size range 1.5-2.5 nm mobility diameter) in Fig. S2 in the Supplement.

Another difference between the instruments is the treatment of the sample relative humidity. In the DMA-train, the aerosol sheath flow is dried by silica gel achieving a relative humidity measured at the sheath inlet of the DMA below 5% for all

experiments in this study. The nano-SMPS uses a water trap to keep the relative humidity of the DMA sheath flow below 20 % during the reported experiments. The Caltech nano-radial DMA, the NAIS and the particle size magnifier used in Lehtipalo et al. (2016) do not deploy any humidity conditioning for the sheath or sample flow, except for the possible decrease in relative humidity as a result of a temperature increase between measurement device and chamber. This effect occurred to some extend for all instruments, even if the sampling lines were insulated. The effect of aerosol dehydration during the measurement is

usually described by the hygroscopic growth factor gf, relating measured diameter $d_{p,m}$ to the actual diameter $d_p$ via $d_p =$ gf $\cdot\, d_{p,m}$.

From the measured aerosol size-distributions we inferred particle growth rates with two complementary methods in order to limit systematic biases in the analysis. In the first method, particle growth rates were measured with the appearance time method, requiring a growing particle population, which can be clearly identified (Dada et al., 2020; Lehtipalo et al., 2014;

Stolzenburg et al., 2018). Fig. S1d in the Supplement demonstrates how the signal in each size channel is fitted by an empirical sigmoidal shape curve estimating the time where 50 % of the maximum signal intensity is reached. These appearance times are fitted with a linear function over the size intervals 1.8-3.2 nm and 3.2-8 nm, with the slope yielding an average growth rate over the interval, shown in Fig. S1b. In the second method, we applied the size- and time-resolving growth rate analysis method INSIDE (Pichelstorfer et al., 2018) to cross-check our results. The INSIDE method uses the measured particle size distribution

at a time $t_1$ and simulates the expected aerosol dynamics (coagulation, wall losses and dilution) until time $t_2$. By comparing to the measured data at $t_2$ and evaluating the general dynamics equation, it infers the condensational growth rate at specified diameters for this time step. The time- and size-resolved growth rates for each experiment were time-averaged for all sizes to yield a statistically more robust result. Compared to the appearance time method, INSIDE requires accurate absolute number size-distributions, while the appearance time method only depends on the relative signal increase. However, INSIDE can confirm the absence of systematic biases like changing precursor vapour concentrations or coagulation and wall loss effects. A combined assessment with both methods should therefore yield a solid estimate of the observed growth rates.

## 2.2 Growth model description

If the evaporation rates of the growing particles are effectively zero due to an extremely low vapour pressure of the condensing vapour, particle growth rates are limited by the collision frequencies of vapour molecules with the growing particles. Our description of particle growth follows the approach of Nieminen et al. (2010), which, in comparison to the equations of mass transfer that can be found in e.g. Seinfeld and Pandis (2016), include the non-negligible effect of vapour molecular size by using a collision frequency between vapour and particle in analogy to coagulation theory (Lehtinen and Kulmala, 2003):

$$GR = \frac{dd_p}{dt} = \frac{\frac{dV_p}{dt}}{\frac{dV_p}{dd_p}} = \frac{k_{coll}(d_v,d_p) \cdot V_v \cdot C_v}{\frac{d}{dd_p}[\frac{\pi}{6}d_p^3]} = \frac{k_{coll}(d_v,d_p) \cdot V_v \cdot C_v}{\pi/2 \cdot d_p^2}, \tag{1}$$

where $d_p$ is the growing particle mass diameter, $V_p$ and $V_v$ are the volume of particle and vapour molecule, $C_v$ is the vapor monomer concentration and $k_{coll}(d_v, d_p)$ is the kinetic collision frequency between particle and vapour. Following Fuchs and Sutugin (1971), the collision frequency for the transition regime is defined by:

$$k_{coll}(d_v, d_p) = 2\pi \cdot (d_v + d_p) \cdot (D_v + D_p) \cdot \frac{1+Kn}{1+(0.377+\frac{4}{3\alpha})Kn+\frac{4}{3\alpha}Kn^2}, \tag{2}$$

where, according to Lehtinen and Kulmala (2003), Knudsen number $Kn$ and mean free path $\lambda$ need to be specified as $Kn = 2\lambda \cdot (d_v + d_p)^{-1}$ and $\lambda = 3(D_v + D_p) \cdot (\bar{c}_v^2 + \bar{c}_p^2)^{-1/2}$, which depend on the diameters $d_{v/p}$ the masses $m_{v/p}$ (within the calculation of the mean thermal velocities $\bar{c}_{v/p}$) and the diffusion coefficients $D_{v/p}$ of the colliding vapour molecules or particles, respectively. Assuming the accommodation coefficient $\alpha$ is unity and relating the volume $V_v$ of the condensing monomer to its molecular mass and (bulk) density $V_V = m_v/\rho_v$, Eq. (1) and (2) determine the hard-sphere kinetic limit for particle growth.

We then additionally consider a collision enhancement of neutral vapour monomers and particles due to attractive van-der-Waals forces, where the collision frequency can be described according to Sceats (1989):

$$k_{coll}(d_v, d_p) = k_K \cdot \left( \sqrt{1 + \left(\frac{k_K}{2k_D}\right)^2} - \left(\frac{k_K}{2k_D}\right) \right), \tag{3}$$

with and the enhanced collision frequency for the continuum regime:

$$k_D = 2\pi \cdot (d_v + d_p) \cdot (D_v + D_p) \cdot E(0) \tag{4}$$

and the enhanced collision frequency for the kinetic regime:

$$k_K = \frac{\pi}{4} \cdot (d_v + d_p)^2 \cdot \left(\frac{8kT}{\pi}\right)^{1/2} \cdot \left(\frac{1}{m_v} + \frac{1}{m_p}\right)^{1/2} \cdot E(\infty) \tag{5}$$

Eq. (3) is designed such that it reaches the correct limits of the free molecular and diffusion regime comparable to the approach of Fuchs and Sutugin (1971), i.e. Eq. (2). However, it includes collision enhancement factors $E(\infty)$ and $E(0)$. These factors can be linked to the attractive potential of van-der-Waals forces. For the continuum regime, this is done by solving the integral:

$$E(0) = \left[\int_{(r_v+r_p)}^{\infty} \left(\frac{r_v+r_p}{x^2}\right) \exp\left(\frac{\phi(x)}{kT}\right) dx\right]^{-1} \tag{6}$$

where $x$ is the relative distance between the centres of the two colliding entities and $\phi(x)$ is the van-der-Waals potential (Hamaker, 1937), which is expressed in terms of the vapour and particle radii $r_{v/p}$:

$$\frac{\phi(x)}{kT} = -\frac{1}{6}\frac{A}{kT}\left(\frac{2\,r_v r_p}{x^2-(r_v+r_p)^2} + \frac{2\,r_v r_p}{x^2-(r_v-r_p)^2} + \ln\left(\frac{x^2-(r_v+r_p)^2}{x^2-(r_v-r_p)^2}\right)\right) \tag{7}$$

Chan and Mozurkewich (2001) provide a fit to the numerical solution of the numerically evaluated integral from Sceats (1989):

$$E(0) = 1 + a_1 \cdot \ln(1 + A') + a_2 \cdot \ln^3(1 + A') , \tag{8}$$

where $a_n$ are the fit parameters and $A'$ is the reduced Hamaker constant, which relates to the Hamaker constant $A$ by $A' = 4A \cdot k^{-1}T^{-1} \cdot d_v d_p \cdot (d_v + d_p)^{-2}$ (Chan and Mozurkewich, 2001; Hamaker, 1937). However, the measurements of this study are conducted completely in the free molecular regime, and hence the derivation of the continuum case will not significantly affect our results. For the free molecular regime enhancement factor $E(\infty)$, an overview of its relation to the Hamaker constant is given in Ouyang et al. (2012). Chan and Mozurkewich (2001) also here used a fit to the solution from Sceats (1989) with the fit parameters $b_n$:

$$E(\infty) = 1 + \frac{\sqrt{A'/3}}{1+b_0\sqrt{A'}} + b_1 \cdot \ln(1 + A') + b_2 \cdot \ln^3(1 + A') , \tag{9}$$

In this study, we compare the results of Sceats (1989), who used Brownian coagulation to describe the collisions, to the simple ballistics approach of Fuchs and Sutugin (1965). There, the minimum distance $x_{\min}$ along the trajectory of two colliding particles with impact parameter $b$ is calculated from conservation of angular momentum und energy:

$$b = x_{\min}\sqrt{1 + \left(\frac{2|\phi(x_{min})|}{\mu v^2}\right)} \tag{10}$$

where $\phi$ is the interaction potential, $\mu$ the reduced mass of the colliding entities and $v$ their relative speed. The critical impact parameter $b_{crit}$ is obtained as the minimum value of $b$ for which the minimum distance still takes a real value larger than $(r_v + r_p)$. The enhancement factor is than related to the critical impact parameter $b_{crit}$ :

$$E(\infty) = \frac{4\,b_{crit}^2}{(d_v+d_p)^2}\sqrt{\frac{3}{2}} \tag{11}$$

Note, that this approach is oversimplified, as the initial velocity of the colliding entities is assumed to be fixed but should actually follow a (Maxwell-Boltzmann) distribution. Ouyang et al. (2012) however concluded that the difference in the derived Hamaker constant is almost negligible.

Using the description of an enhanced collision kernel, the particle growth rates measured with the DMA-train can be fitted with the Hamaker constant as the single free parameter of the fit. As the theoretical growth rates are compared to appearance

time growth rates, which are measured as a time difference in signal appearance $\Delta t$ over a certain size-interval $\Delta d_p$ (ranging from $d_{\text{init}}$ to $d_{\text{final}}$), a comparison with experimental values requires integration of Eq. (1):

$$GR\,(d_{\text{init}}, d_{\text{final}}) = \frac{\Delta d_p}{\Delta t} = (d_{\text{final}} - d_{\text{init}})\Big/ \int_{d_{\text{init}}}^{d_{\text{final}}} \frac{\pi/2 \cdot d_p^2}{k_{coll}(d_v, d_p) \cdot V_v \cdot C_v}\, \mathrm{d}d_p, \tag{12}$$

Eq. (12) includes several properties of the condensing vapour and the growing particles. Sulphuric acid molecules are usually hydrated at typical ambient relative humidity. While the thermodynamic model E-AIM (Wexler et al., 2002) predicts on

average 2 water molecules attached to a sulphuric acid monomer at 298 K and 40-60% relative humidity, quantum chemical studies predict 1-2 water molecules average hydration for these conditions (Henschel et al., 2014; Kurtén et al., 2007; Temelso et al., 2012). Moreover, also the hydration state of the particles in the chamber is not directly measured and might be altered during the sampling process, which requires information on the hygroscopic growth factor (see Section 2.1).

We examine the effect of hydration using three different approaches: In a first naïve approach we assume that no dehydration

occurs during measurement and the particle sulphuric acid mass fraction is equal to the vapour mass fraction, i.e. $w = M_{H2SO4}/m_v$, with $m_v = M_{H2SO4} + 2M_{H2O}$ (assuming 2 water molecules attached to the sulphuric acid monomer), where $M_{H2SO4}$ and $M_{H2O}$ are the molecular mass of sulphuric acid and water, respectively. In the second approach, we assume a dry measurement, and in this case the growth of the measured dry particles is described by uptake of sulphuric monomers only, i.e. $m_v = M_{H2SO4}$. However, for the actual vapour and particle size used in the collision kernel $k_{coll}(d_v, d_p)$ the

hydrated sizes are used. We again assume an average hydration for the monomer with 2 water molecules as above and an average hygroscopic growth factor of 1.25 for all particle sizes and RH values of our experiments. The latter is an average value of the results of Biskos et al. (2009) for highly acidic sulphuric acid sub-10 nm particles at 40-60 % relative humidity. In the third approach, we take into account that the extent of hydration might vary with size and relative humidity. We use modelled composition data from MABNAG (Yli-Juuti et al., 2013) in order to predict the sulfuric acid mass fraction $w(RH, T)$

(see Fig S4a in the Supplement) and calculate the hygroscopic growth factor:

$$\mathrm{gf} = \left( \frac{w(RH_m, T_m) \cdot \rho_{\text{sol}}(w(RH_m, T_m), T_m)}{w(RH, T) \cdot \rho_{\text{sol}}(w(RH, T), T)} \right)^{1/3}, \tag{13}$$

where $\rho_{\text{sol}}$ is a parametrization of the density of the sulphuric acid water solution (Myhre et al., 1998) and $w(RH, T)$ and $w(RH_m, T_m)$ are the mass fractions of sulphuric acid in the growing and measured particles, respectively. We follow the considerations of Verheggen and Mozurkewich (2002), in order to separate growth by sulphuric acid addition and water uptake

by differentiating the hydrated particle volume $V_p = m_{H2SO4}/w\rho_{sol}$. Both, the numerator (particle sulphuric acid mass $m_{H2SO4}$) and denominator (sulphuric acid mass fraction and solution density) depend on time. The addition of sulphuric acid is again described in analogy to coagulation theory, resulting in:

$$\frac{\pi}{2} d_p^2 \frac{\mathrm{d}dp}{\mathrm{d}t} = \frac{k_{coll}(d_v, d_p) \cdot m_v \cdot C_v}{w \cdot \rho} - \frac{\pi d_p^3}{6} \frac{\mathrm{d}\ln(w\rho)}{\mathrm{d}t} = \frac{k_{coll}(d_v, d_p) \cdot m_v \cdot C_v}{w \cdot \rho} - \frac{\pi d_p^3}{6} \frac{\mathrm{d}\ln(w\rho)}{\mathrm{d}d_p} \frac{\mathrm{d}d_p}{\mathrm{d}t} \tag{15}$$

Eq. (14) contains a first term for addition of pure sulphuric acid and a second term for water uptake. It can be solved for the particle growth rate $\frac{ddp}{dt}$:

$$GR = \frac{2 \cdot k_{coll}(d_v, d_p) \cdot m_v \cdot C_v}{w(RH,T) \cdot \rho(RH,T) \cdot \pi \cdot d_p^2 \cdot \left(1 + \frac{d_p}{3} \cdot \frac{dln(w\rho)}{ddp}\right)}, \tag{15}$$

In this case, we assume $m_v = M_{H2SO4}$, but use the hydrated monomer diameter $d_v$ in the collision kernel. For the particles we now use the hydrated size, i.e. $d_p = \text{gf} \cdot d_{p,m}$ with gf and $w(RH,T)$ now taken from the model. We compare the MABNAG predictions in Fig. S4b in the Supplement to SAWNUC (Ehrhart et al., 2016), which takes into account only sulphuric acid and water, while MABNAG also includes ammonia. MABNAG predicts a significantly lower water content at larger sizes (>2.5 nm) even at 3 pptv ammonia. In addition, previous experiments in the CLOUD chamber suggested that even background level ammonia has an influence on the hygroscopic growth factor (Kim et al., 2016), similar to Biskos et al. (2009) also indicating some extend of neutralization for sub-10 nm particles at low ammonia. Due to these presumably better prediction of the particle hydration by MABNAG for sizes larger than 2.5 nm, we choose the results of Fig. S4a in the Supplement even if it might overestimate the hydration at small sizes. We neglected the effect of ammonia addition upon collisions in all three approaches so far, but test the assumption $m_v = M_{H2SO4} + 2M_{H2O} + 1M_{NH3}$ together with different vapour hydrations in our systematic uncertainties estimate in Fig. S5 in the Supplement. All used parameters for vapour and particles for all approaches are summarized in Table S1 in the Supplement.

## 2.3 Global model description

We implement the results of our growth-rate measurements for sulphuric acid driven growth in a global model (Mann et al., 2010; Mulcahy et al., 2018), which includes sulphuric acid-water binary nucleation. However, the model does not include ternary nucleation schemes (Dunne et al., 2016) and pure biogenic nucleation (Gordon et al., 2016) and will therefore underestimate the impact of nucleation on the global aerosol and CCN budget. Here as a baseline case we use the geometric hard-spheres kinetic growth rate based on bulk-density (Eq. (3)) and compare this to the collision enhanced growth (Eq. (4)-(9)). In the model, growth between the nucleation size and 3 nm is treated with the equation of Kerminen and Kulmala (Kerminen and Kulmala, 2002), which gives the fraction of particles surviving to 3 nm at a given growth and loss rate. Here as a baseline case we use the geometric hard-sphere kinetic growth rate based on bulk-density (Eq. (3)) and compare this to the collision enhanced growth (Eq. (4)-(9)). For larger sizes, aerosol growth in the model is calculated by solving the condensation equations. Therefore no direct growth parametrization can be altered, but as condensational growth scales linearly with the diffusion coefficient of the condensing vapour, we increased sulphuric acid diffusion for condensation in the nucleation mode (2-10 nm) and in the Aitken mode (10-100 nm). The enhancement factors are derived for the median diameters of the modes (7.6 and 57 nm respectively) at cloud base level (1 km). However, this constant factor of increase in diffusion coefficient, and hence flux onto particles, for all particles of the entire mode, might underestimate the impact of the collision

enhancement. Rapid growth is increasingly important for the smallest particles, which actually have a higher collision

enhancement compared to particles with the size of the mode median diameters.

## 3 Results

### 3.1 Collision enhancement

Figure 1 shows the particle growth rates for two size-intervals (Fig. 1a, 1.8-3.2 nm mobility diameter and Fig. 1b, 3.2-8.0 nm mobility diameter) versus the sulphuric acid monomer concentration, correlating linearly. No significant dependencies on

temperature, ionization levels in the chamber or the concentration of ammonia are evident. While the effect of temperature expected from theory is small and cannot be discerned within the statistical uncertainties of our measurements (Nieminen et al., 2010), the insignificant influence of ammonia and ionization level on the growth rate differs from previous findings (Lehtipalo et al., 2016).

We compare the measured growth rates of this study with the results from Lehtipalo et al. (2016) in Fig. S2 in the Supplement. In contrast to our results, elevated ammonia (~ 1000 pptv) led to increased growth rates in that study. The major difference is the narrower size range for the growth-rate measurements (1.5-2.5 nm mobility diameter) due to a different set of instrumentation. For smaller sizes and at low ammonia, sulphuric acid evaporation likely plays a role due to an increased Kelvin term. The stabilizing effect of ammonia is certainly relevant at the sizes of the nucleating clusters (Kirkby et al., 2011).

For our results, we confirm the absence of significant evaporation rates above 2 nm by an independent experiment presented in Fig. 2. It demonstrates that, in the absence of gas-phase sulphuric acid, the coagulation and dilution corrected loss rates of particles ($k_{tot}^{meas} - k_{dil} - k_{coag}^{avg}$) over all sizes follow the expected size-dependence of wall losses which is inferred from the sulphuric acid monomer decay. Evaporation would cause another term distorting the balance equation (also depending on the relative abundances of the particles during the decay), causing a deviation from the expected wall loss rate.

The insignificant effect of ammonia on growth (Fig. 1) and the same high ratio (>100, Fig. S3a in the Supplement) between sulphuric acid monomer and dimer concentrations for all experiments, point towards a negligible influence of clustering on our measured growth rates (Li and McMurry, 2018). Moreover, in Fig. S3b in the Supplement, we show with a model including sulphuric acid/ammonia clustering and evaporation, that no cluster contribution is indeed expected even at elevated ammonia concentrations (Kürten, 2019).

In the absence of evaporation and strong clustering, our growth-rate data provide a direct measurement of the condensational growth at the kinetic limit caused by sulphuric acid monomers only. We find the measured growth rates both with and without addition of ammonia to be significantly above the geometric hard-sphere limit (Eq. (1)-(2)) of kinetic condensation (Nieminen et al., 2010). For this comparison we followed a naïve approach, assuming an average hydration of the monomer by 2 water molecules and applied the resulting mass fraction to find the bulk density (Myhre et al., 1998). The observed enhancement is

similar to Lehtipalo et al. (2016) in the case when evaporation was suppressed by ammonia (see Fig. S2). We also measure a

growth-rate enhancement for the larger size range (Fig. 1b), which should be less sensitive to evaporation. The faster growth rates might be due to an enhanced collision frequency, which can be attributed to van-der-Waals forces, either permanent dipole-(induced) dipole interactions between polar sulphuric acid molecules and particles or London dispersion forces (London, 1937). The magnitude of the enhancement is described by the Hamaker constant $A$ (Hamaker, 1937), which we use as the single free parameter to fit a collision enhanced kinetic limit . For the Brownian coagulation model linking the Hamaker constant to the collision kernel, i.e. Eq. (4)-(9) (Sceats, 1989), we find $A = (4.6 \pm 1.5 \, (stat.)) \cdot 10^{-20}$ J. If we apply a ballistics approach in the free molecular regime (Fuchs and Sutugin, 1965; Ouyang et al., 2012), we derive a slightly higher value of $A = 8.7 \cdot 10^{-20}$ J, but both yield comparable values to previous results (Chan and Mozurkewich, 2001; McMurry, 1980).

An enhancement due to charge-dipole interactions between the polar sulphuric acid monomers and charged particles is not significant in our total (neutral plus charged particle) growth rate measurements, as shown in Fig. 1, where we observe no difference between growth rates under neutral and galactic cosmic ray ionization levels. From average-dipole-orientation theory (Su and Bowers, 1973), a small enhancement is expected in collision frequency for charged particles above 2 nm (Nadykto and Yu, 2003), which should affect the growth rate (Laakso et al., 2003; Lehtipalo et al., 2016). We find an enhancement factor of 1.45 by comparing the total to the ion growth rate as shown in Fig. 3, which is in good agreement with theory. However, the total growth rate is influenced on a minor level by the faster ion growth because at the representative galactic cosmic ray ionization levels and sulphuric acid concentrations in our experiments, most (more than 75%) of the growing particles are neutral (see Fig. 3). However, effects of ion condensation and charge-dipole enhancement might be stronger at lower sulphuric acid concentrations (Svensmark et al., 2017).

### 3.2 Size-dependency and hydration effects

Condensational growth at the geometric kinetic limit predicts increasing growth rates with decreasing particle sizes due to the non-negligible effect of vapour molecule size on the collision cross-section (Nieminen et al., 2010), which, was not yet shown experimentally. Furthermore, the collision enhancement due to van-der-Waals forces and the collision enhancement due to dipole-charge interactions also depend on the comparative size of the condensing vapour and the growing particle. Fig. 4a illustrates the theoretical predictions of the size-dependency of the collision rate of sulphuric acid monomers with larger particles, including van-der-Waals forces and dipole-charge interactions. The enhancement factor compared to the hard-sphere kinetic limit is shown for both the Brownian coagulation model (Sceats, 1989) and the ballistics approach (Fuchs and Sutugin, 1965) (2.1 and 2.3 for the free molecular regime, respectively) and is comparable to previous experimental results (Kürten et al., 2014; Lehtipalo et al., 2016) and quantum chemical calculations (Halonen et al., 2019).

Besides the approach for calculating the kinetic enhancement factor, also the description of particle hydration might play a crucial role. Up to now, we used the naïve assumption that vapour and particle hydration are the same and that particles are measured at their hydrated size. However, during sampling the measured particles are potentially dried. To investigate the

effect of particle hydration, we use the DMA-train data of Fig. 1 to fit the collision enhancement for two alternative approaches,
one where we assume that particles are measured dry and one where we separate the uptake of water and sulphuric acid
condensation (Verheggen and Mozurkewich, 2002) by using modelled particle composition data from SAWNUC (Ehrhart et
al., 2016) or MABNAG (Yli-Juuti et al., 2013). We compare the predictions for the size-dependency of all approaches with
the measured growth rates of all instruments normalized to $10^7$ cm$^{-3}$ in Fig 4b. In addition, we show the growth rates using the
time- and size-resolving growth rate analysis method INSIDE (Pichelstorfer et al., 2018), which agrees with the appearance
time method demonstrating a minor systematic bias in our growth rate determination. All approaches reproduce the size-
dependency on an acceptable level ($R^2$ larger than 0.87). The separation approach yields higher growth rates at the smallest
sizes due to the overestimation of hydration by MABNAG below 2.5 nm. For SAWNUC composition data, which presumably
describe the cluster hydration better, the $R^2$ is however only 0.66 not reproducing the observed size-dependency. This is
possibly caused by the assumed too high hydration for larger sizes. The simple dry measurement approach might thus be a
good approximation to the predictions of both MABNAG and SAWNUC for the size range of interest (see Fig. S4b in the
Supplement). We estimate the systematic uncertainty of the results in Fig. S5 in the Supplement, also including the effects of
different vapour hydration, ammonia addition and sulphuric acid measurement uncertainty. All approaches overlap largely
within their systematic uncertainties with $A = \left(5.2^{+9.7}_{-3.4}(\mathrm{syst.})\right) \cdot 10^{-20}$ J as the best estimate of a combined assessment
(assuming the Brownian coagulation model). We also give a first order approximation to our measured growth rates and their
size-dependency for the conditions of our experiments:

$$GR(\mathrm{nm\ h^{-1}}) = \left[2.68 \cdot d_p(\mathrm{nm})^{-1.27} + 0.81\right] \cdot [\mathrm{H_2SO_4(cm^{-3})} \cdot 10^{-7}] \tag{16}$$

### 3.3 Global implications

The observed steep increase of the growth rates with decreasing size shows that the collision enhancement due to van-der-
Waals forces is especially important for the smallest particles. As these are the most vulnerable for losses to pre-existing
aerosols, their survival probability in the atmosphere is directly affected, altering the CCN budget (Pierce and Adams, 2007)
or promoting new particle formation in urban environments (Kulmala et al., 2017). In order to test the effects of collision
enhancement in sulphuric acid growth on a global scale, we use the atmosphere-only configuration of the United Kingdom
Earth System Model (UKESM1) (Mulcahy et al., 2018; Walters et al., 2019) which includes the GLOMAP aerosol
microphysics module describing nucleation and growth (Mann et al., 2010). Figure 5 illustrates the global model results
comparing the baseline case (no collision enhancement) with a collision enhancement simulation (with enhancement factors
of 2.2, 1.8 and 1.3 for cluster, nucleation and Aitken mode) for the present-day atmosphere. The absolute particle number
concentrations averaged over all longitudes are shown in Figure 5a, indicating changes of more than 50%, especially at high
altitudes (>10 km; Figure 5b), where most aerosol particles originate from pure sulphuric-acid driven NPF. The importance of
the nucleation process, and therefore the growth-rate enhancement, is lower at lower altitudes and in the northern hemisphere,
mainly due to the higher condensation sink and the restriction of the model to only sulphuric acid-water binary nucleation.

However, the significant enhancement of sulphuric acid driven nanoparticle growth in the upper troposphere may be important in quantifying sources of stratospheric aerosols and cirrus cloud condensation nuclei (Brock et al., 1995; Deshler, 2008) and needs to be accounted for in future model development.

## 4 Discussion

Understanding nanoparticle growth driven by sulphuric acid is extremely important for modelling the present-day atmosphere. Our measured growth rates cover a wide range of representative atmospheric conditions below 20 °C and reveal that sulphuric acid growth proceeds faster than the geometric hard-sphere kinetic limit. Such faster growth rates in the cluster size range could be in part responsible for the occurrence of NPF in polluted environments (Kulmala et al., 2017). Our results suggest that for sizes larger than 2 nm this collision enhancement due to van-der-Waals forces can be more important than dipole-

charge interactions or base-stabilization by ammonia. However, a better knowledge of the chemical composition of the condensing vapour and growing sub-10 nm particles could further improve our understanding of molecular collision rates. For smaller sizes, evaporation of sulphuric acid and charge effects need to be considered, but the size-range covered by our measurements is sufficient for the used global model, which nucleates particles at 1.7 nm. We find significantly increased upper tropospheric aerosol concentrations, but the global impact of van-der-Waals forces in nanoparticle growth might be even

higher due to the model limitations to binary sulphuric-acid water nucleation. Our results should therefore be considered in future model development, especially when discussing the importance of changing sulphuric acid levels due to reduced anthropogenic emissions of $SO_2$. Moreover, our parametrization of pure sulphuric acid growth rates will help to identify the contribution to growth of other co-condensing vapours in ambient and laboratory experiments, as they set a new baseline for kinetic condensation of sulphuric acid. Several simplifications have often been applied to kinetic particle growth, including

hard-spheres collision based on bulk density and neglect of vapour size to the collision cross section; our results provide clear experimental verification that these simplifications are no longer fit for increasingly accurate measurements at these tiny yet critical sizes.

*Data availability:* All presented datasets are available from the corresponding author upon reasonable request.

*Author Contributions*: D.S., M.Sim. A.K., K.L., H.F., X.H, S.Bri., M.X., R.B. A.B., S.Brä., L.C.M., D.C., B.C., A.D., J.Dom., J.Dup., I.E.H., L.F., L.G.C., M.H., C.K., W.K., H.L., C.P.L., M.L., Z.L., V.M., H.E.M., T.M., E.P., J.P., M.P., M.P.R., S.Scho., S.Schu., J.S., M.Sip., G.S., Y.S., Y.J.T., A.T., A.C.W., M.W., Y.W., S.K.W., D.W., P.J.W., Y.W., Q.Y., M.Z.W., U.B., J.C., R.C.F., R.V., J.K., P.M.W. prepared the CLOUD facility or measuring instruments, D.S., M.Sim., A.R., K.L. X.H. S.Bri., M.X., A.A., R.B., A.B, L.B., S.Brä., L.C.M., D.C., L.D., A.D., J.Dup., I.E.H., H.F., L.F., L.G.C., M.H., C.K., T.K.K., W.K., H.L., C.P.L., M.L., Z.L., H.E.M., R.M., T.M., W.N., E.P., J.P., M.P.R., B.R., S.Schu., G.S., C.T., Y.J.T., A.T., M.V.P., A.C.W., M.W., S.K.W., D.W., P.J.W., Y.W., Q.Y., M.Z.W. collected the data, D.S., M.Sim., A.R., H.G., T.N., L.P., L.D., H.F., S.E., M.H., C.K., A.C.W., S.K.W. analysed the data, D.S., M.S., A.R., A.K., K.L., T.N., X.H., M.X., J.Dom., J.Dup., I.E.H., T.K.K., T.P., M.P.R., M.Sip., U.B., K.S.C., J.C., N.M.D., R.C.F., A.H., M.K., J.L., R.V., J.K., P.M.W. were involved in the scientific discussion and interpretation of the data, D.S., A.K., K.L., H.G., N.M.D., J.K., P.M.W. wrote the manuscript.

*Competing interests:* The authors declare no competing financial interests.

*Acknowledgements*: We thank CERN for supporting CLOUD with technical and financial resources, and for providing a particle beam from the CERN Proton Synchrotron. We thank P. Carrie, L.-P. De Menezes, J. Dumollard, K. Ivanova, F. Josa, T.Keber, I. Krasin, R. Kristic, A. Laassiri, O. S. Maksumov, B. Marichy, H. Martinati, S. V. Mizin, R. Sitals, A. Wasem and M. Wilhelmsson for their contributions to the experiment. This research has received funding from the EC Seventh Framework Programme and European Union's Horizon 2020 programme (Marie Skłodowska Curie  no. 764991 "CLOUD-MOTION", MC-COFUND grant no. 665779, ERC projects no. 616075 "NANODYNAMITE", no. 714621 "GASPARCON"), the German Federal Ministry of Education and Research (no. 01LK1601A "CLOUD-16"), the Swiss National Science Foundation (projects no. 200020_152907, 20FI20_159851, 200021_169090, 200020_172602 and 20FI20_172622), the Academy of Finland (projects 296628, 299574, 307331, 310682), the Austrian Science Fund (FWF; project no. J-3951, project no. P27295-N20, project no. J-4241), the Portuguese Foundation for Science and Technology (FCT; project no. CERN/FIS-COM/0014/2017), the U.S. National Science Foundation (grants AGS-1649147, AGS-1801280, AGS-1602086, AGS-1801329).

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

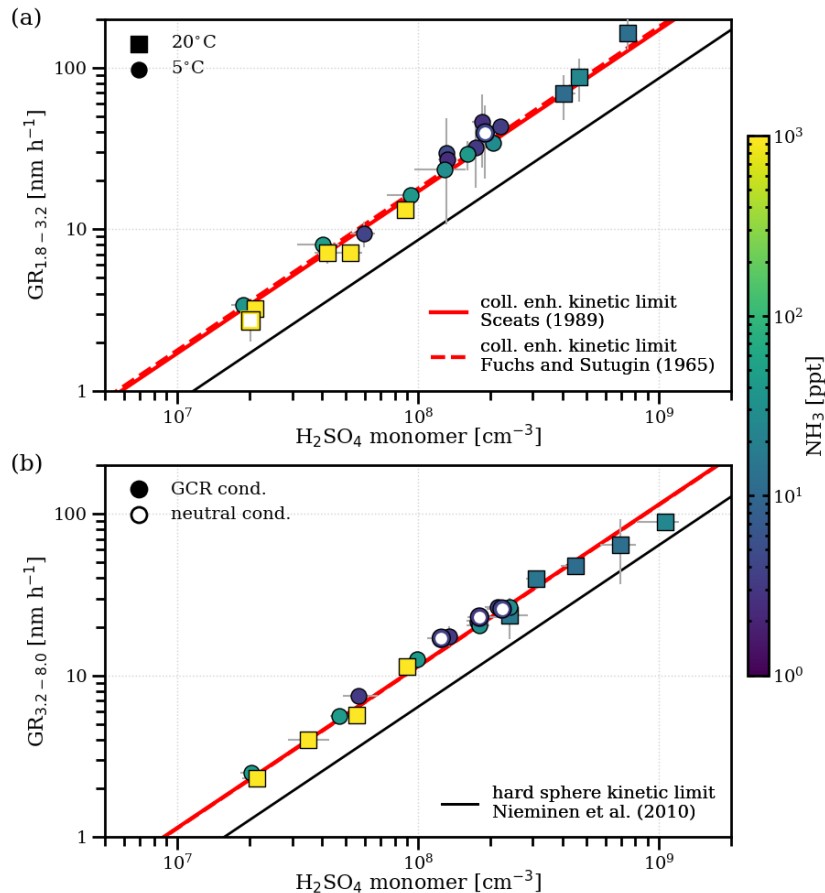

**Figure 1:** Growth rates of nanoparticles in two size-intervals versus measured gas-phase sulphuric acid monomer concentration. (a) shows growth rates for the size-interval between 1.8-3.2 nm (mobility diameter; 1.5-2.9 nm in mass diameter), while (b) shows the growth rates for the size-interval 3.2-8.0 nm (mobility diameter, 2.9-7.7 nm in mass diameter). The colour code represents the measured $NH_3$ concentration during the growth period. Squares are measurements at 20°C, circles at 5°C. Filled symbols represent runs under ambient galactic cosmic ray ionization levels, and open symbols under neutral conditions. Error bars for the data points represent the statistical uncertainty in the appearance time growth rate measurements and the maximum variation of the sulphuric acid concentration during the growth period, also explaining the slight deviations from linearity at high sulphuric acid concentrations, where stable conditions are not fully reached. The black line show the geometric limit of kinetic assuming the same hydration for the condensing cluster and the measured particles (Nieminen et al., 2010). The red solid line shows the fit of Eq. (12) to the data with the Hamaker constant as free parameter assuming a Brownian coagulation model for the enhanced collision kernel (Sceats, 1989), while the red dashed line uses a ballistics approach (Fuchs and Sutugin, 1965).

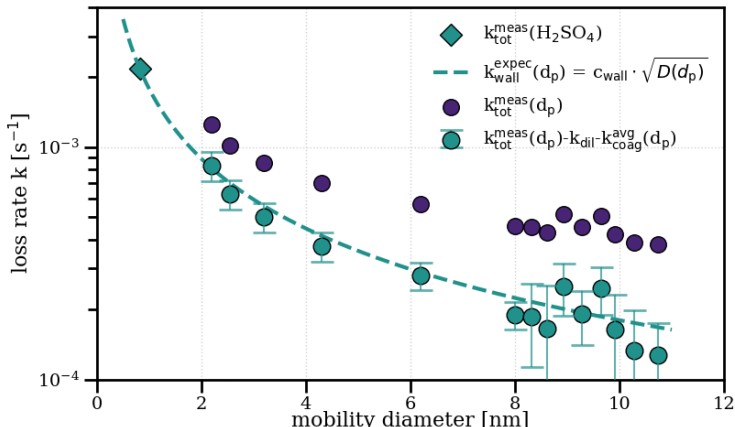

**Figure 2:** Measurement of zero sulphuric acid evaporation rates. Total loss rates of sulphuric acid and ammonia particles with mobility diameter shown on the x-axis measured during a decay experiment (5°C, 60% relative humidity, 1000 pptv $NH_3$), by switching off the UV lights after a particle growth stage, which stops the production of sulphuric acid and subsequently nucleation and growth. After sulphuric acid is reduced to background level, the exponential decay rate of the remaining particles in the chamber is measured ($k_{tot}^{meas}$, blue circles), which was not possible for the 1.8 nm channel due to low statistics. Decay of particles in the chamber is dominated by wall loss, dilution loss and coagulation loss to other particles. Particle loss rates are corrected for an averaged coagulation loss during the decay ($k_{coag}^{avg}$) to all particles larger than $d_p$ and for the dilution loss ($k_{dil}$) (turquoise circles). They agree well with the expected wall loss rate $k_{wall}(d_p) = C_{wall} \cdot \sqrt{D_p(d_p)}$ (red dashed line) with $C_{wall} = 0.0077$ s$^{-0.5}$ cm$^{-1}$ inferred from an independent sulphuric acid decay experiment in the absence of a particle sink, where the mobility diameter is assumed to be 0.82 nm (Ehrhart et al., 2016) (turquoise diamond). This suggests that there is negligible evaporation from the sulphuric acid particles above ca. 2 nm under the above mentioned experimental conditions, which would introduce another term disturbing the balance equation at each size. As all our growth-rate measurements, independent of the ammonia concentration and temperature, fall on the same line (see Fig. 1), this also points towards negligible evaporation effects at reduced ammonia levels (below 10 pptv) and up to 20°C.

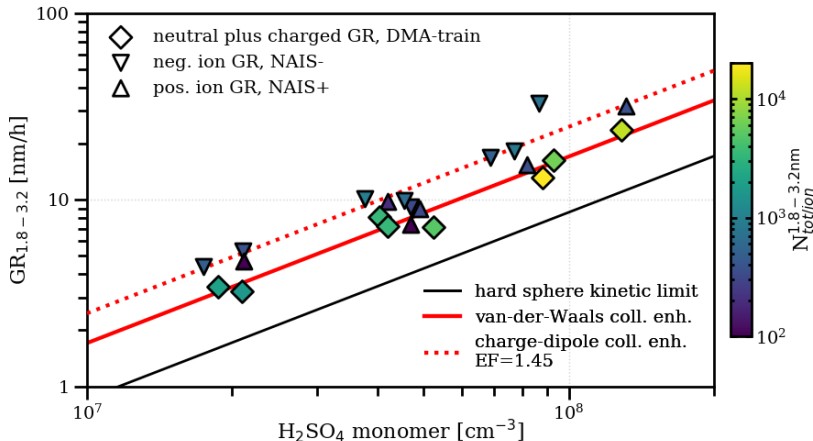

**Figure 3:** The effect of charge on growth. Measured growth rates of 1.8-3.2 nm (mobility diameter) particles and ions in experiments with ammonia above 25 pptv. The DMA-train measures both neutral and charged particles (diamonds) whereas the NAIS+/- (Manninen et al., 2009) measures purely charged particles (triangles). Both, the positively and negatively charged particle population have a faster apparent growth rate than the total particle population due to an enhanced collision rate from dipole-charge interactions. We measure a multiplicative charge enhancement factor of 1.45 in this size range with a combined fit to both polarities (red dotted line), which is consistent with estimates

from average dipole orientation theory (Nadykto and Yu, 2003). At galactic cosmic rays ionization levels in the chamber, the charged fraction of the growing particles in the size-range 1.8-3.2 nm (mobility diameter) is between 5 and 25%. This is demonstrated by the colour code which indicates the integrated total or ion number concentration over the growth rate size interval averaged during the growth period. The fit of the appearance time for the total particle population is therefore affected on a minor level by the small earlier appearing charged fraction.

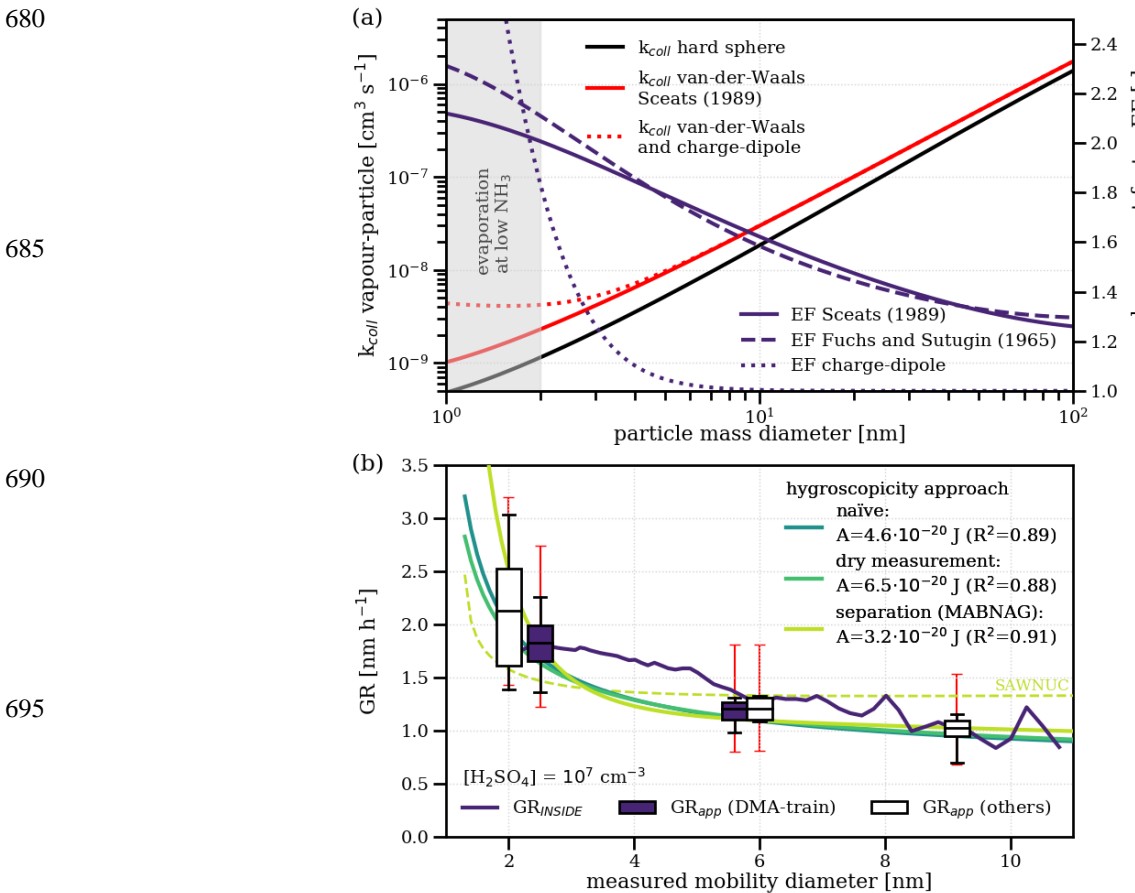

**Figure 4:** The size-dependency of sulphuric acid growth. (a) shows the theoretical collision rate of hydrated sulphuric acid vapour molecules ($m_v = M_{H2SO4} + 2M_{H2O}$) with particles of a certain mass diameter. The black line represents the hard-sphere limit, the red solid line also includes a collision enhancement due to van der Waals forces based on the approach of ($A = 4.6 \cdot 10^{-20}$ J), and the red dashed line on the approach of ($A = 8.7 \cdot 10^{-20}$ J). The red dotted line additionally includes charge-dipole interactions based on average-dipole-orientation theory. The blue lines show the enhancement factor of a single attractive force compared to the hard-sphere limit. (b) shows the measured size-dependency of growth rates normalized to a sulphuric acid concentration of $10^7$ cm$^{-3}$. The solid blue line shows the growth rates inferred with the INSIDE method. Filled boxes represent the appearance time growth rates from the DMA-train used to fit the Hamaker constant. Empty boxes represent appearance time growth rates from other instruments including the results from Lehtipalo et al. (2016) with high (>100 pptv) NH$_3$ concentrations. The boxes indicate the median and the 50% interquartile range of the data, while the whiskers represent the 90% quantile. The red small errorbars indicate the -33%/+50% systematic uncertainty in the sulphuric acid measurement. We show the size-dependency of three different approaches for particle hygroscopicity. The naïve approach (solid turquoise line), assuming the same hydration for vapour and particle; the dry measurement approach (solid light green line), assuming that the DMA-train measures completely dehydrated particles; and the separation approach (solid yellow line), assuming that available composition data from MABNAG can disentangle water uptake from sulphuric acid condensation. The separation approach using SAWNUC composition data is also shown as a dashed yellow line.

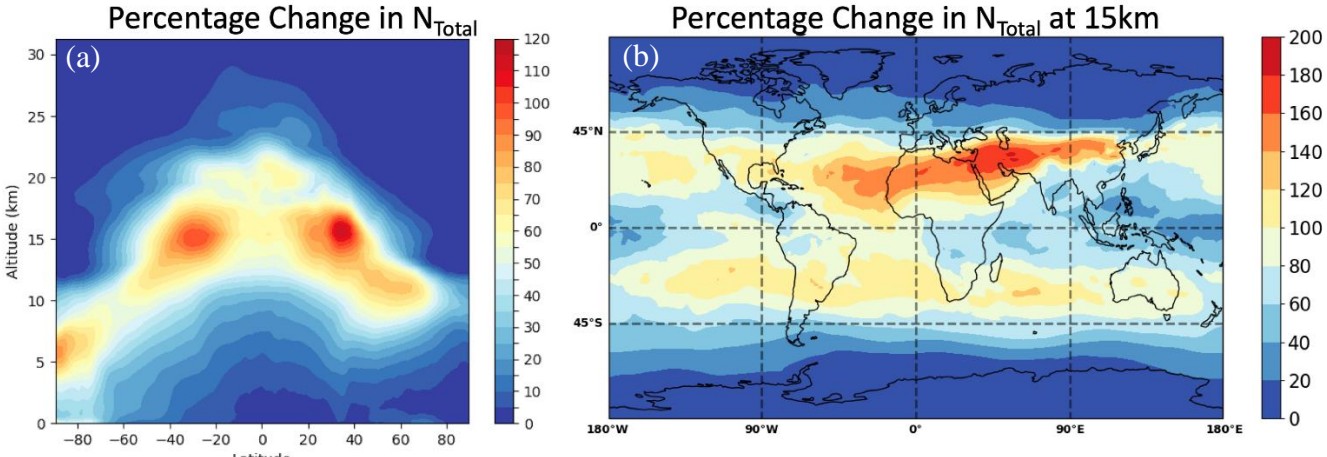

715 **Figure 5:** Increased global aerosol number concentrations due to the collision enhancement. Results from a global modeling study of the present-day atmosphere. (a) shows the relative change in total aerosol number concentration (particles larger 3 nm) averaged over all longitudes in a vertical profile if a collision enhancement is considered in sulphuric acid growth. (b) shows the relative increase at 15 km altitude on a global scale where the effects are most significant. Higher relative changes would be expected also at lower altitudes, if the model is adjusted for ternary sulphuric acid-water-ammonia nucleation.