# Peer review of "Enhanced growth rate of atmospheric particles from sulphuric acid"

_Atmospheric Chemistry and Physics, 2019_

## Referee Comment (RC1) · Anonymous Referee #2 · 5 Dec 2019

This is important work and therefore needs the best, possible analysis and also a clear presentation of uncertainties. I plan to submit comments on the presentation style and the context of the work at a later date.

(0) Are the particle sizes listed as mobility diameters (see Larriba et al. 2011)? Assumed to be the case for the rest of these comments. . .

0.a) Assuming ammonia is not affecting composition significantly, relative humidities of 38 and 60 % gives bulk densities times wt. fraction of 0.67 and 0.49, respectively. This indicates an expected 30 % difference in growth rates for these two conditions. Is there a humidity dependence in the GRs?

(1) Potential bias in the measurements at the smallest sizes. As an example, the 1.8

nm mobility diameter channel corresponds to a 1.5 nm mass diameter.

1.a) If we assume they are large enough to attain bulk properties (and ammonia has no effect) there are about 6 sulfuric acid molecules in this DMA-train channel. Early literature results (Froyd and Lovejoy, Hogan lab) indicate that negatively charged 4 acid clusters can hold very little water thus these small particles will shrink upon charging (since it is likely that 6 acid clusters will also lose much of their water content upon charging.) There is a potential bias in what size to assign to the detection at these small channels: they may actually be a little larger in size than 1.8 mobility diameter before charging. Is it known at what size shrinkage becomes negligible?

1.b) On the other hand, the 1.8-2.2 nm DMA-train signals may come from neutral particles that have not attained enough water to reach bulk values. Yet there will be a size at which bulk properties are better achieved, say 3.2 nm for argument. Then between the small-diameter channels and the 3.2 nm diameter (or larger) there will be swelling due to water uptake. This is an effect that is not directly related to collisional rate coefficients.

These two effects would affect the measured GRs in the same direction it seems.

(2) Confusion about Figure 2. It seems that the monomer is plotted with mass diameter (assuming bulk hydration and density) while the DMA-train results are plotted as mobility diameters? Secondly, where is the 1.8 nm data? This size might be especially sensitive to evaporation. Thirdly: it is not clear whether an additional first-order loss term due to evaporation would appear as a negative or positive deviation from the sum of the calculated/estimated first-order loss rates (i.e. wall loss, coagulation, and dilution): for example, a 2.55 nm particle evaporating might cause a bump up in the 2.2 nm channel's signal (depending on relative abundances etc.)

(3) The smallest channels' appearance times may be most affected by the time it takes $H_2SO_4$ to reach steady state. How has this been addressed?

(4) Are not the GRs subject to a -50/+33 % systematic uncertainty due to the +50/-33% uncertainty in measured H2SO4? This should be reflected in the figures and particularly in Fig. 4b where it is stated that systematic uncertainties are represented by the pink band.

---

## Short Comment (SC1) · 5 Dec 2019

I am commenting on this manuscript because I think the equations presented and analysis methods utilized in this work require greater explanation and clarification. I believe there are a combination of typographical errors (potentiall) and conceptual issues to address in the analysis.

1. First, it seems the authors equate the mobility diameter inferred from DMA measurements without correcting for the size of the gas molecule (or if they are correcting for gas molecule physical size, they are not stating this). As per Larriba et al (2011, doi: 10.1080/02786826.2010.546820), the physical diameter of a particle in air is ~0.3 nm smaller than the mobility diameter. If using mobility diameter for collision rate calculations, this lack of correction could have a huge consequence at smaller sizes (e.g. $(2.8\ nm/2.5\ nm)^2 = 1.25$, a 25% difference in hard sphere predicted growth rate). This seems that it would ultimately effect the collision rate calculated, and interpretation of the Hamaker constant.

2. Equation S1 has a factor of 2 in the numerator that is either not correct or not fully justified. The authors state that "$k_{coll}$ is the kinetic collision frequency between particle and vapor, which is accounted for twice to include collisions in both ways." What does "both ways" mean? Why would something grow at twice the condensation rate?

3. I think equation S2 is not correctly written, or at least, it has the wrong limits. When the ratio $k_D/k_K$ (Knudsen number for collisions) is large, then it does correctly converge to $k_K$. However, it appears that as this ratio becomes very small, equation S2 would approach zero, when it needs to converge exactly to $k_D$ to be an accurate transition regime kernel. In Chan and Mozurkewich (equations 27 and 28 of their paper), the do provide different expressions than what is given here. However, these equations would also have the same issue, (an incorrect limit). The authors should look at the transition regime expressions of Fuchs or Dahneke (or more recent developments) to see how the transition regime equation must be constructed to converge to the correct limits.

4. The authors appear to be using equations from Chan and Mozurkewich which have typographical errors in the original manuscript and are copied here. Just below equation (S2), the author note that the kinetic regime rate coefficient $k_K$ is given by the expression:

$$k_K = \frac{\pi}{8}(d_p + d_v)^2 \sqrt{\frac{8kT}{\pi m_{p,v}}} E(\infty)$$

However, $k_K$ should be the product of the projected area of the combined particle-vapor molecule at contact, their relative mean thermal speed, and an enhancement factor (Chan and Mozurkewich have the same typographical error). Therefore it should be:

$$k_K = \frac{\pi}{4}(d_p + d_v)^2 \sqrt{\frac{8kT}{\pi m_{p,v}}} E(\infty)$$

Like the gas molecule correction, this change will affect the inferred Hamaker constant. At the same time, oddly enough, as written, the authors' extra factor of 2.0 and their extra division by

2.0 would cancel out in the free molecular regime. However, then their equation for $k_D$ would not be correct with the factor of 2.0 in equation S1 included.

5. Aside from the issues with equation S2, use of the equations of Chan and Mozurkewich alone, which come from the theoretical derivations of Sceats, may not be accurate. More recent theoretical investigations of van der Waals enhancement in collision rates (doi: 10.1063/1.4742064) suggest that for a given Hamaker constant, Sceats overestimates the enhancement factor in the free molecular limit. The noted reference discusses more tractable approaches which agree with collision rates from trajectory calculations.

6. Equations S4 for E(0). I think the authors should show that this functional form follows exactly from the Fuchs integral for the enhancement factor in the continuum/diffusive regime, which is an exact integral and typically easily calculable for a given potential interaction. Similarly, it is not clear that equation S3 follows from analyzing ballistic regime in the appropriate manner. The authors should be aware that the equations in Chan and Mozurkewich do not appear in the original Sceats paper, as far as I can tell they are a regression fit to results in Sceats's work (from his plots).

7. It would seem more reasonable for the authors to use a different approach either in lieu of or in addition to the equations of Chan and Mozurkewish, i.e. for the authors to either compare to the equations of Ouyang et al (doi: 10.1063/1.4742064) and/or of Fuchs & Sutugin (doi: simple but accurate: 10.1016/0095-8522(65)90031-0). In the free molecular limit, Ouyang et al follow Fuchs's approach but integrate across the velocity distribution, while Fuchs assumes a single approach speed.

---

## Author Comment (AC1) · 13 Dec 2019

**Response to Short Comment from Christopher Hogan on "Enhanced Growth Rate of Atmospheric Particles from Sulfuric Acid"**

We thank Chris Hogan for thoroughly checking our manuscript on potential errors in the analysis. This will certainly help improving the quality and validity of the presented results.

*First, it seems the authors equate the mobility diameter inferred from DMA measurements without correcting for the size of the gas molecule (or if they are correcting for gas molecule physical size, they are not stating this). As per Larriba et al (2011, doi: 10.1080/02786826.2010.546820), the physical diameter of a particle in air is ~0.3 nm smaller than the mobility diameter.*

Indeed we used mobility diameter for our growth rate calculations, and collision kernels are typically given for mass diameters, which according to Larriba et al. (2011) are approximately 0.3 nm smaller. We appreciate that you pointed that out which will make our analysis certainly more correct. We will revise that throughout the manuscript and always specify which definition of diameter is used. In fact, we find a Hamaker constant of $5.3 \times 10^{-20}$ J with the adjusted size range which is 20 % lower than before. Nevertheless, this is still within the general uncertainty of our Hamaker constant estimate and still in good agreement with earlier results. As can be seen from the below Figure (updated Figure 4, with geom. Diameter on x-axis), the effect of faster GRs than the hard sphere assumption is still significant.

[Figure]

*Equation S1 has a factor of 2 in the numerator that is either not correct or not fully justified. The authors state that "$k_{coll}$ is the kinetic collision frequency between particle and vapor, which is accounted for twice to include collisions in both ways." What does "both ways" mean? Why would something grow at twice the condensation rate? The authors appear to be using equations from Chan and Mozurkewich which have typographical errors in the original manuscript and are copied here.*

Thanks for pointing this out, indeed a very good remark. This factor of 2 is indeed connected to the factor of 2 which appears in the collision rate and results from a wrong interpretation by the authors. As already noted the factors do cancel out, so there is no effect on the inferred GR. Apparently, we

were following Chan & Mozurkewich (2001) too strictly in our derivation. We will correct for that in the revised manuscript, update our calculations and delete the sentence on the counting of the collisions.

The GR will be defined as:

$$GR = \frac{k_{coll} \cdot m_v \cdot C_v}{\rho \frac{\pi}{2} d_p^2} \tag{S1new}$$

*I think equation S2 is not correctly written, or at least, it has the wrong limits. When the ratio $k_D/k_k$ (Knudsen number for collisions) is large, then it does correctly converge to $k_k$. However, it appears that as this ratio becomes very small, equation S2 would approach zero, when it needs to converge exactly to $k_d$ to be an accurate transition regime kernel. In Chan and Mozurkewich (equations 27 and 28 of their paper), the do provide different expressions than what is given here.*

This seems to be a misunderstanding. We provide the exact same expression as Chan & Mozurkewich (2001), just the factor of 2 in the collision rate for the continuum regime is placed at a different position:

| Term | Discussion paper | Chan & Mozurkewich with $a_{i/j} = d_{v/p}/2$ | Revised version |
|------|------------------|-----------------------------------------------|-----------------|
| $k_k$ | $\frac{\pi}{8}(d_v + d_p)(\bar{c}_v^2 + \bar{c}_p^2)^{1/2} E(\infty)$ | $\frac{\pi}{8}(d_v + d_p)(\bar{c}_v^2 + \bar{c}_p^2)^{1/2} E(\infty)$ | $\frac{\pi}{4}(d_v + d_p)(\bar{c}_v^2 + \bar{c}_p^2)^{1/2} E(\infty)$ |
| $k_D$ | $2\pi(D_v + D_p)(d_v + d_p)E(0)$ | $\pi(D_v + D_p)(d_v + d_p)E(0)$ | $2\pi(D_v + D_p)(d_v + d_p)E(0)$ |
| $k_T$ | $k_k \cdot \left( \sqrt{1 + \left(\frac{k_k}{k_D}\right)^2} - \left(\frac{k_k}{k_D}\right) \right)$ | $k_k \cdot \left( \sqrt{1 + \left(\frac{k_k}{2k_D}\right)^2} - \left(\frac{k_k}{2k_D}\right) \right)$ | $k_k \cdot \left( \sqrt{1 + \left(\frac{k_k}{2k_D}\right)^2} - \left(\frac{k_k}{2k_D}\right) \right)$ |

*However, these equations would also have the same issue, (an incorrect limit). The authors should look at the transition regime expressions of Fuchs or Dahneke (or more recent developments) to see how the transition regime equation must be constructed to converge to the correct limits.*

First, we want to point out that the formulation of the transition regime correction is in any case not dramatically affecting our results as we were analysing sub-10 nm growth rates, which should be entirely dominated by the free molecular regime. On line 128 in the main text we underline that already. However, we are also confused by the statement of the wrong limits. Also Sceats (1987) used a similar expression and claims it reaches the correct limit (Eq. 7 in Sceats work). For the case considered in our manuscript (fixed vapor size and varying particle size) we also find that the limits are correct. This can be seen from the below Figure (similar to Figure 4 a in the current manuscript), which shows the collision kernel of a sulfuric acid molecule with a particle of diameter $d_p$. Both the transition regime expression by Fuchs & Sutugin (1971) and the one used by Chan & Mozurkewich (2001) are approaching the correct limits and yield quite similar results. Note, that we adjusted the expression of Chan & Mozurkewich by the wrong factor of 2 as shown in the above Table under "Revised version". We therefore see no further reason to fundamentally change our analysis, but we will once again cross-check all our calculations and equations before submitting a revised version of our manuscript.

[Figure]

*Aside from the issues with equation S2, use of the equations of Chan and Mozurkewich alone, which come from the theoretical derivations of Sceats, may not be accurate. More recent theoretical investigations of van der Waals enhancement in collision rates (doi: 10.1063/1.4742064) suggest that for a given Hamaker constant, Sceats overestimates the enhancement factor in the free molecular limit. The noted reference discusses more tractable approaches which agree with collision rates from trajectory calculations.*

We appreciate this comment pointing us towards other approaches in literature. We will try to make use of them in the revised version of the manuscript, to crosscheck our analysis.

*Equations S4 for E(0). I think the authors should show that this functional form follows exactly from the Fuchs integral for the enhancement factor in the continuum/diffusive regime, which is an exact integral and typically easily calculable for a given potential interaction. Similarly, it is not clear that equation S3 follows from analyzing ballistic regime in the appropriate manner. The authors should be aware that the equations in Chan and Mozurkewich do not appear in the original Sceats paper, as far as I can tell they are a regression fit to results in Sceats's work (from his plots).*

We agree with Chris Hogan, that the manuscript needs some deeper explanations on how the work of Chan & Mozurkewich (2001) relates to other work and we will clarify some of the derivations in the revised version of the manuscript.

*It would seem more reasonable for the authors to use a different approach either in lieu of or in addition to the equations of Chan and Mozurkewish, i.e. for the authors to either compare to the equations of Ouyang et al (doi: 10.1063/1.4742064) and/or of Fuchs & Sutugin (doi: simple but accurate: 10.1016/0095-8522(65)90031-0). In the free molecular limit, Ouyang et al follow Fuchs's approach but integrate across the velocity distribution, while Fuchs assumes a single approach speed.*

We understand that the Chan & Mozurkewich (2001) approach might be oversimplified and that more recent developments on that exist. As already mentioned above, Chan & Mozurkewich (2001) was used as the blueprint for this analysis and we might have followed it too rigorously. Adding another approach will certainly help improving the quality of the manuscript and will be done in the revised version of the manuscript. However, we also want to point out that these different derivations will only influence the result of the inferred Hamaker constant, which is anyways subject to systematic uncertainties which are difficult to reduce with the chosen approach. We will comment this point separately in our reply to Ref.2. In this context, we want to emphasize that this manuscript is about the observation of growth rates faster than expected from a hard sphere assumption under pure, well-controlled and atmospherically relevant conditions, which seem to have a global impact. This was also the reason to choose Atmos. Chem. Phys. for dissemination instead of a journal which focusses more on pure Physical Chemistry and collision kinetics.

---

## Referee Comment (RC2) · Anonymous Referee #2 · 4 Jan 2020

l 77. " A Debye-type permanent .... " is introduced here but not referenced and not explained. l 122 also.

l 98. " unmeasureable " does not seem to be the correct word here, suggesting that in principle no one is able to do such a measurement.

l 110. One would like to see the pH2SO4 from E-AIM as a reference point here...

l 111. monomer to dimer ratio: is it shown in this paper? Is there a reference?

l 119. Assuming both the monomer and cluster attain bulk composition and densities ?

l 137. the line is 1.45 times the lower line. Yet this does not seem to be a fit as there are 7 pts that are clearly below the line and only two pts that are clearly above

[Figure]

the line (negative and positive are different?) Please explain. Also, dipole moments, polarizabilities for theory? It might be useful to have a table somewhere listing all the molecular and cluster parameters that are relevant (a few select clusters.)

l 138. barely influenced? This phrase should be replaced by an actual number/upper limit.

l 146. Earlier papers have suggested that the Hamaker effect depends mostly on comparative sizes, not on the absolute size.

l 153. Too strongly worded: both could be biased or the underlying measurements or assumptions could be wrong.

l 157. The effect of ions is most here? Possible bias then in the GRs for 1.8 to 3.2 nm ?

l 162. Should state that in principle, their ought to be a humidity dependence...

l 185. Too strong. A clear demonstration of no effect due to NH3-stabilization needs to be put in the context of binary evaporation rates (e.g. E-AIM): do these clusters even need a base to avoid significant evaporation? On the other hand, there may be a low-level diamine or some other strong nucleator, which I think has not been ruled out for the warm CLOUD experiments. Furthermore, the systematic uncertainty in SA leads to a factor of ∼two error bars...

---

## Referee Comment (RC3) · Anonymous Referee #3 · 6 Jan 2020

This manuscript (acp-2019-755) describes measurements of <10-nm diameter nanoparticle growth rates from sulfuric acid nucleation under well-controlled, low-ammonia conditions in the CLOUD chamber. Analysis of the growth rates indicates that nanoparticle growth proceeds at a rate faster than the hard-sphere collision rate for sulfuric acid condensation. The explanation for this enhanced growth rate is due to an enhancement in the sulfuric acid collision rate from dipole-induced dipole interactions between the vapor molecules and nanoparticles. Consequently, smaller particles exhibit enhanced growth rates relative to larger particles. Incorporation of these enhanced growth rates for the smallest particles into a global model increases the predicted particle number concentration by 50% in the upper free troposphere, demonstrating a potential global impact.

[Figure]

This is a well-written manuscript describing important results. The manuscript is within the scope of Atmospheric Chemistry and Physics, and will be suitable for publication once the comments below, along with those of Anonymous Referee 2 and Christopher Hogan, are fully addressed.

Comments:

1. Both Anonymous Referee 2 and Christopher Hogan raise significant concerns about the diameter values (mobility vs. mass) used in this study and how those values impact the experimental analysis and conclusions. This reviewer agrees with these comments but will not reproduce them here. The comments of Anonymous Referee 2 and Chris Hogan must both be addressed in revision. Changing the particle diameter used in the analysis may also impact the comparison to Lehtipalo (2016).

2. This manuscript would benefit from providing additional context with respect to reconciling measured nanoparticle growth rates to gas phase sulfuric acid concentrations. Significant effort was devoted to developing and applying models that provide closure for nanoparticulate and gas phase sulfuric acid (see work from McMurry, Smith, and Johnston, e.g. Kuang et al., 2010, 10.5194/acp-10-8469-2010; Smith et al., 2008, 10.1029/2007GL032523; Bzdek et al., 2013, 10.1039/C3FD00039G). While these studies looked at somewhat larger particles ($\sim$5-20 nm diameter) during field studies, the main conclusion was that sulfuric acid addition to ambient nanoparticles proceeds approximately at the collision limited rate.

3. Figure 1b: What is the cause of the deviation from linearity at high [H2SO4]?

4. Figure 4: The explanation of the enhancement factor for the charge-dipole interactions (blue dotted line) should be clarified. The caption states that the blue lines are enhancements relative to the hard-sphere limit (black dotted line). If that is the case, then would not the blue-dotted line be obtained by dividing the red dotted line by the black dotted line in Fig. 4a? Instead, it appears the blue dotted line arises from dividing the red dotted line by the red solid line (dipole-induced dipole interaction).

5. Figure S1c: Is there a factor of several orders of magnitude (e.g. 107) missing from the axis label for sulfuric acid?

6. Figure S3: Please clarify what is meant by "the effect of hidden sulfuric acid".

7. Figure S4: Are the labels on this figure correct? The figure caption states that the yellow bars indicate water content, but the label in the figure only concerns sulfuric acid.

---

## Short Comment (SC2) · 8 Jan 2020

We also agree with the comments made by Chris Hogan and we would like to put some emphasis on possible issues in the theoretical part of this relevant study.

In the manuscript the Hamaker constant for sulfuric acid particles is calculated from experimental growth rates. However, the selected collision/coagulation rate model by Chan and Mozurkewich (2001) should be discussed in more detail as the estimated magnitude of the long-range interaction i.e. the Hamaker constant follows from it. The model has been used previously for collisions of molecules with atmospherical relevance in the free molecular regime (e.g. Kürten et al. 2014, doi:10.1073/pnas.1404853111; Lehtipalo et al. 2016, doi:10.1038/ncomms11594;

[Figure]

Kürten et al. 2018, doi:10.5194/acp-18-845-2018) even though the model is based on Brownian coagulation in a field of force (Sceats 1986, doi:10.1063/1.450636) which does not describe the correct transport physics of collisions of molecules in the gas phase (Ouyang et al. 2012, doi:10.1063/1.4742064).

In a very recent study, the rate enhancement in collision of two sulfuric acid molecules was estimated using atomistic trajectory simulations and theoretical approaches (Halonen et al. 2019, doi:10.5194/acp-19-13355-2019). In this particular case, the Sceats model is significantly overestimating (about 40%) the rate coefficient at 300 K in the free molecular regime. We are agreeing with Chris Hogan, who in his comment suggested that different approaches should be concidered along with the one by Chan and Mozurkewich. (Furthermore the study by Halonen et al. suggests that a simple Langevin-like model is performing better especially if the Keesom interaction is negligible).

Also, as it is often repeated in the manuscript, the claim that the enhancement is solely due to Debye-type permanent dipole-induced dipole interaction should be rationalized somewhere. For a single sulfuric acid molecule and a spherical particle containing a large number of molecules, neglecting the Keesom interactions is a good approximation, however for the interaction of a single molecule and a small cluster this may not be the case. And in any case, the London dispersion forces are always present.

---

## Author Response (AR1)

**Response to the Anonymous Referees and Short Comments given during the interactive discussion of the manuscript "Enhanced growth rate of atmospheric particles from sulfuric acid"**

**Response to Anonymous Referee #2:**

*This is important work and therefore needs the best, possible analysis and also a clear presentation of uncertainties. I plan to submit comments on the presentation style and the context of the work at a later date.*

We thank the Reviewer for carefully checking our manuscript and pointing us towards some important improvements of the presented analysis. Below we give a point-by-point response including the changes we made to the revised version of the manuscript.

*(0) Are the particle sizes listed as mobility diameters (see Larriba et al. 2011)? Assumed to be the case for the rest of these comments…*

As already reported in the author comment to Christopher Hogan, this is indeed a major point, which we did not consider in the initial manuscript. We repeated our analysis using mass diameters (-0.3 nm from mobility diameters according to Larriba et al. (2011)) and updated the following figures: Figure 2 (x-axis gives mobility diameter and the sulfuric acid monomer is plotted as mobility diameter, see below), Figure 4 b (mass diameter now). Figure 1, 3 and S2 will still give mobility diameters (which we specify in the Figure caption), because they show our direct measurement results which are measured as mobility diameters. We also changed the corresponding Figure captions and added a sentence on **lines 120-123** of the revised manuscript text and adjusted the Supplement. This should clarify the points raised by Christopher Hogan and both reviewers. We also updated our calculations accordingly (see also next comments) and present a newly derived Hamaker constant on **line 132** of the revised manuscript.

*0.a) Assuming ammonia is not affecting composition significantly, relative humidities of 38 and 60 % gives bulk densities times wt. fraction of 0.67 and 0.49, respectively. This indicates an expected 30 % difference in growth rates for these two conditions. Is there a humidity dependence in the GRs?*

*(1) Potential bias in the measurements at the smallest sizes. As an example, the 1.8 nm mobility diameter channel corresponds to a 1.5 nm mass diameter.*

*1.a) If we assume they are large enough to attain bulk properties (and ammonia has no effect) there are about 6 sulfuric acid molecules in this DMA-train channel. Early literature results (Froyd and Lovejoy, Hogan lab) indicate that negatively charged 4 acid clusters can hold very little water thus these small particles will shrink upon charging (since it is likely that 6 acid clusters will also lose much of their water content upon charging.) There is a potential bias in what size to assign to the detection at these small channels: they may actually be a little larger in size than 1.8 mobility diameter before charging. Is it known at what size shrinkage becomes negligible?*

*1.b) On the other hand, the 1.8-2.2 nm DMA-train signals may come from neutral particles that have not attained enough water to reach bulk values. Yet there will be a size at which bulk properties are better achieved, say 3.2 nm for argument. Then be-tween the small-diameter channels and the 3.2 nm diameter (or larger) there will be swelling due to water uptake. This is an effect that is not directly related to collisional rate coefficients.*

*These two effects would affect the measured GRs in the same direction it seems.*

We thank the reviewer for drawing our attention more onto the effects of humidity, hydration and charging. We agree that the effect of water should be discussed more thoroughly in the manuscript. We added a sentence on **lines 120-123** in the revised manuscript, pointing towards the basic assumptions on vapor and particle hydration. Moreover, we adjusted the sentence on **lines 135-137** in the revised manuscript relating our results to the uncertainties in aerosol water content.

First, it is important to note that we measure the dry particle diameter with our measurement set-up. For the DMA-train we achieved a relative humidity below 5% in the sheath-flow of all channels. Similar to the above comment, this means that the measured diameter (even if corrected to mass diameter) cannot be used in the calculations of the collision kernel, but assumptions on hydration for both the vapor and the particle have to be made, which we did not consider so far in our derivations. We therefore updated our calculations and the corresponding text in the Supplement.

The average hydration of the sulfuric acid monomer can be assumed to be on average 1-2 water molecules (as previously). This is supported by Henschel et al. (2014) and also E-AIM, both are now cited. For the growing particles, Ehrhart et al. (2016) showed that the average hydration is fairly constant for particles between 1.8 and 10 nm in mobility diameter resulting in a sulfuric acid mass fraction of 0.5-0.6 (Figure S3, Ehrhart et al. (2016)). In addition, classical thermodynamics including a Kelvin effect, predict a similar ratio (0.6) for 298.15 K and 50 % relative

humidity (see Seinfeld and Pandis 2012). We therefore applied the SAWNUC code to predict the most probable water content of the particles and clusters for the temperatures and relative humidities in the manuscript. The most probable water content can be used to calculate the sulphuric acid mass fraction, which can then be related to a hygroscopic growth factor. Altogether, we thus derive the particle size relevant for the collision kinetics and can derive the Hamaker constant. We also included this water dependency now in our systematic uncertainty estimate. We show the SAWNUC results in a new panel added to Figure S4.

With our derivations and calculations updated, we can clarify the reviewer's raised concerns. A change between 38% and 60% relative humidity does neither significantly change the hydration of the monomer (see Henschel et al., 2014), nor the mass fraction of sulfuric acid on a level that it could be discerned within the uncertainties of our measurements. This is demonstrated in the added panel of Figure 4a and the updated uncertainty analysis shown in Figure 4b. The additional uncertainty in mass fraction of ±0.1 leads to minor differences in the Hamaker constant. The problem due to change in hydration upon charging effect should not be present above 1.8 nm in mobility diameter as shown by Ehrhart et al. (2016). Last, the effect of particle swelling is now fully taken into account by the size-dependent hydration derived from SAWNUC.

*(2) Confusion about Figure 2. It seems that the monomer is plotted with mass diameter (assuming bulk hydration and density) while the DMA-train results are plotted as mobility diameters? Secondly, where is the 1.8 nm data? This size might be especially sensitive to evaporation. Thirdly: it is not clear whether an additional first-order loss term due to evaporation would appear as a negative or positive deviation from the sum of the calculated/estimated first-order loss rates (i.e. wall loss, coagulation, and dilution): for example, a 2.55 nm particle evaporating might cause a bump up in the 2.2nm channel's signal (depending on relative abundances etc.)*

The sulfuric acid monomer was already plotted as mobility diameter here, using earlier results on the mobility of sulfuric acid (Ehrhart et al., 2016). We added the reference in the Figure caption. The 1.8 nm was not possible to fit with an exponential decay, because the counting statistics were too low after the sulfuric acid vapour was down to background level. We added this also in the Figure caption. The reviewer is correct that the effect of evaporation on the loss rates depends on the relative abundances of the particles at each size, but also on the size-dependency of the evaporation rates $\gamma_i$. For any other case than $\gamma_i = 0 \; \forall \; i$, the extra term to the balance equation would only vanish if but $\gamma_i N_i = \gamma_{i+1} N_{i+1}$ during the entire decay. In all other cases a distortion to the measured decay rates would appear. We thus modified the paragraph on **lines 107-110**.

*(3) The smallest channels' appearance times may be most affected by the time it takes H2SO4 to reach steady state. How has this been addressed?*

We now plot Figure 1 with the maximum and minimum sulfuric acid concentration measured during the time period where growth was observed in the corresponding interval. As can be seen, from the updated Figure 1, steady-state H2SO4 concentrations are reached during almost all experiments. During experiments with very high H2SO4 concentrations, the concentration dropped significantly due to the buildup of a large condensation sink during the run. Therefore, we did find some larger errors for these measurements, also explaining the deviation of these measurement points from the linear sulfuric acid-growth rate relation (see also answer to Referee #3).

*(4) Are not the GRs subject to a -50/+33 % systematic uncertainty due to the +50/-33% uncertainty in measured H2SO4? This should be reflected in the figures and particularly in Fig. 4b where it is stated that systematic uncertainties are represented by the pink band.*

The reviewer is correct, that due to the linear relationship between H2SO4 and GRs, the error can be directly translated to a -50/+33 % uncertainty in the growth rate measurement. We therefore updated Figure 4b with additional errorbars indicating the systematic uncertainty of the sulfuric acid measurements. Figure 1 should still present our direct measurement results (see above) and will not show any systematic uncertainties in order to keep the clarity of the Figure.

*l 77. " A Debye-type permanent .... " is introduced here but not referenced and not explained. l 122 also.*

This comment is also related to the short comment from Roope Halonen raised in the interactive discussion. We agree with both Referee #2 and Roope Halonen, that the van-der-Waals term described here is not soley due to Debye-type forces, but also other interactions could be significant. In the revised manuscript, we therefore refrain from calling the interaction Debye-type forces but use the term van-der-Waals forces only, and added a specifying sentence on **line 127** of the revised manuscript.

*l 98. "unmeasureable " does not seem to be the correct word here, suggesting that in principle no one is able to do such a measurement.*

We corrected this in the revised version to insignificant.

*l 111. monomer to dimer ratio: is it shown in this paper? Is there a reference?*

The number comes from our own measurements. We decided to add another panel to Figure S3 in the Supplement, demonstrating that the monomer/dimer ratio remains constant upon the addition of ammonia at 5 deg C.

*l 119. Assuming both the monomer and cluster attain bulk composition and densities?*
We added another sub sentence here, concretizing our assumptions on vapor and particle hydration and refer the reader to the detailed derivations in the Supplement.

*l 137. the line is 1.45 times the lower line. Yet this does not seem to be a fit as there are 7 pts that are clearly below the line and only two pts that are clearly above the line (negative and positive are different?) Please explain. Also, dipole moments, polarizabilities for theory? It might be useful to have a table somewhere listing all the molecular and cluster parameters that are relevant (a few select clusters.)*
The line is a fit where the multiplicative factor is the free parameter. We fit both polarities together as we have limited statistics. It indeed seems as there could be a slight difference between the polarities, but due to the limited amount of measurement points we decided to not speculate too much. The overall charge enhancement however is significant. We added a sentence in the Figure caption to clarify that we fit both polarities combined. We also thank the reviewer for requesting a Table with all the important parameters of the paper, which will help to reproduce our results and derivations. We added this to the Supplement and refer to it in the main text **line 124**.

*l 138. barely influenced? This phrase should be replaced by an actual number/upper limit.*
The influence should be indeed negligible as we are determining the appearance time at 50% of the plateau concentration. If at maximum only 25% of the total particle population grow by ion-dipole enhancement the 50% appearance time will still identify the neutral particle population reaching this size. The slightly skewed leading edge could still lead to minor differences. However, a numerical estimation for the appearance time error and also the expected minor influence on the growth rate would require size-distribution modelling, which we do not consider necessary for the statement that the impact is rather minor. We adjusted the sentence slightly on **line 148** in the revised manuscript. In the Figure caption we also changed the last sentence slightly.

*l 146. Earlier papers have suggested that the Hamaker effect depends mostly on comparative sizes, not on the absolute size.*
Changed to "(…) depend on the comparative size of condensing vapor and growing particle." on **line 156** in the revised manuscript.

*l 153. Too strongly worded: both could be biased or the underlying measurements or assumptions could be wrong.*
This sentence should only clarify that the used methods do agree and that no biases exist in the approach of determining growth rates. We do not want to express with this sentence that there are no systematic effects biasing our entire analysis. We therefore slightly modified the sentence on **line 163** in the revised manuscript.

*l 157. The effect of ions is most here? Possible bias then in the GRs for 1.8 to 3.2 nm?*
As explained above, the low fraction of charged particles in our experiments prevents any major bias from ions, also for the size range between 1.8 and 3.2 nm. Moreover, both our measured growth rates and the calculations take into account that the growth rate measurement is an integrated over a certain size-interval. Therefore, effects which only affect one size at a minor level cannot be identified within the uncertainties of our measurements nor can they significantly disturb the calculation of the enhancement.

*l 162. Should state that in principle, their ought to be a humidity dependence...*
We agree with reviewer that this parametrization might not be complete, but still be useful for models and quick calculations in order to avoid the entire Hamaker formalism. We clarified this, but moved the entire parametrization to the Supplement.

*l 185. Too strong. A clear demonstration of no effect due to NH3-stabilization needs to be put in the context of binary evaporation rates (e.g. E-AIM): do these clusters even need a base to avoid significant evaporation? On the other hand, there may be a low level diamine or some other strong nucleator, which I think has not been ruled out for the warm CLOUD experiments. Furthermore, the systematic uncertainty in SA leads to a factor of two error bars...*
We changed the wording to "our results suggest" and "can be more important" on **lines 189-191** in the revised manuscript.

**Response to Anonymous Referee #3:**

*This manuscript (acp-2019-755) describes measurements of <10-nm diameter nanoparticle growth rates from sulfuric acid nucleation under well-controlled, low-ammonia conditions in the CLOUD chamber. Analysis of the growth rates indicates that nanoparticle growth proceeds at a rate faster than the hard-sphere collision rate for sulfuric acid condensation. The explanation for this enhanced growth rate is due to an enhancement in the sulfuric acid collision rate from dipole-induced dipole inter-actions between the vapor molecules and nanoparticles. Consequently, smaller particles exhibit enhanced growth rates relative to larger particles. Incorporation of these enhanced growth rates for the smallest particles into a global model increases the predicted particle number concentration by 50% in the upper free troposphere, demonstrating a potential global impact.*

*This is a well-written manuscript describing important results. The manuscript is within the scope of Atmospheric Chemistry and Physics, and will be suitable for publication once the comments below, along with those of Anonymous Referee 2 and Christopher Hogan, are fully addressed.*

We thank the Referee for thoughtfully reviewing our manuscript, which certainly helped to clarify the message of the paper. Please find detailed point-by-point responses including the changes made in the manuscript.

*1. Both Anonymous Referee 2 and Christopher Hogan raise significant concerns about the diameter values (mobility vs. mass) used in this study and how those values impact the experimental analysis and conclusions. This reviewer agrees with these comments but will not reproduce them here. The comments of Anonymous Referee 2 and Chris Hogan must both be addressed in revision. Changing the particle diameter used in the analysis may also impact the comparison to Lehtipalo (2016).*

We agree with all referees that these points have to be clarified in the revised manuscript. Please find the point-by-point replies in the comments to Anonymous Referee #2.

*2. This manuscript would benefit from providing additional context with respect to reconciling measured nanoparticle growth rates to gas phase sulfuric acid concentrations. Significant effort was devoted to developing and applying models that provide closure for nanoparticulate and gas phase sulfuric acid (see work from McMurry, Smith, and Johnston, e.g. Kuang et al., 2010, 10.5194/acp-10-8469-2010; Smith et al., 2008, 10.1029/2007GL032523; Bzdek et al., 2013, 10.1039/C3FD00039G). While these studies looked at somewhat larger particles (∼5-20 nm diameter) during field studies, the main conclusion was that sulfuric acid addition to ambient nanoparticles proceeds approximately at the collision limited rate.*

We agree with Referee #3 that some additional context on already existing field measurements comparing growth with the hard-sphere approximation for sulfuric acid will benefit the introduction of the manuscript. We therefore added some of the mentioned references and modified the text in **line 81** of the revised manuscript.

*3. Figure 1b: What is the cause of the deviation from linearity at high [H2SO4]?*

We agree with Referee #3 that this is worth a sentence of explanation in the text, so we added another sentence in the Figure caption (see also the comments to Referee #2): "(…) maximum variation of the sulfuric acid concentration during the growth period in the corresponding interval measurement, also explaining the slight deviations at high sulfuric acid concentrations, where stable conditions are not fully reached"

*4. Figure 4: The explanation of the enhancement factor for the charge-dipole interactions (blue dotted line) should be clarified. The caption states that the blue lines are enhancements relative to the hard-sphere limit (black dotted line). If that is the case, then would not the blue-dotted line be obtained by dividing the red dotted line by the black dotted line in Fig. 4a? Instead, it appears the blue dotted line arises from dividing the red dotted line by the red solid line (dipole-induced dipole interaction).*

We agree with the referee that we should resolve this confusion on how the enhancement factors are defined. Both enhancement factors show the individual enhancement of a single additional attractive interaction (either the charge-dipole or the van-der-Waals forces), so while the solid blue line is indeed the solid red line divided by the black dashed line, the dashed blue line represents the and tried to clarify this in the Figure caption: "The blue lines show the enhancement factor of a single attractive force (charge interaction or van-der-Waals forces) compared to the hard-sphere limit."

*5. Figure S1c: Is there a factor of several orders of magnitude (e.g. 107) missing from the axis label for sulfuric acid?*

The order of magnitude is actually already displayed on the top of the axis. We enlarged this label and changed it to $10^8$ instead of 1e8 in the revised version of the manuscript.

*6. Figure S3: Please clarify what is meant by "the effect of hidden sulfuric acid".*

We clarified this in the beginning of the modified Figure caption.

*7. Figure S4: Are the labels on this figure correct? The figure caption states that the yellow bars indicate water content, but the label in the figure only concerns sulfuric acid.*

Thanks to the referee for pointing that out, indeed the legend has a typo. We corrected that in the revised version of the manuscript.

**Response to Roope Halonen, Short Comment posted on ACPD January 8, 2020**

*We also agree with the comments made by Chris Hogan and we would like to put some emphasis on possible issues in the theoretical part of this relevant study. In the manuscript, the Hamaker constant for sulfuric acid particles is calculated from experimental growth rates. However, the selected collision/coagulation rate model by Chan and Mozurkewich (2001) should be discussed in more detail as the estimated magnitude of the long-range interaction i.e. the Hamaker constant follows from it. The model has been used previously for collisions of molecules with atmospherical relevance in the free molecular regime (e.g. Kürten et*

*al.2014,doi:10.1073/pnas.1404853111; Lehtipalo et al. 2016, doi:10.1038/ncomms11594; Kürten et al. 2018, doi:10.5194/acp-18-845-2018) even though the model is based on Brownian coagulation in a field of force (Sceats 1986, doi:10.1063/1.450636) which does not describe the correct transport physics of collisions of molecules in the gas phase (Ouyang et al. 2012, doi:10.1063/1.4742064). In a very recent study, the rate enhancement in collision of two sulfuric acid molecules was estimated using atomistic trajectory simulations and theoretical approaches (Halonen et al. 2019, doi:10.5194/acp-19-13355-2019). In this particular case, the Sceats model is significantly overestimating (about 40%) the rate coefficient at 300 K in the free molecular regime. We are agreeing with Chris Hogan, who in his comment suggested that different approaches should be considered along with the one by Chan and Mozurkewich. (Furthermore the study by Halonen et al. suggests that a simple Langevin-like model is performing better especially if the Keesom interaction is negligible).*

We agree with Roope Halonen and Chris Hogan that a more thorough discussion of the different approaches is necessary. As mentionned in the reply to Chris Hogan short comment, we will discuss the different derivations of the enhancement factor and its link to the Hamaker constant in more detail now in the Supplement. We also did calculate the Hamaker constant using the approach of Fuchs and Sutugin in order to demonstrate the difference in theoretical approach and added a new sentence on **lines 134-135**. We than added the Fuchs and Sutugin size-dependence also to Figure 4a and b and added a sentence on **line 165-166**. Additionally, we refer to the study of Halonen et al., who actually inferred an enhancement factor, which is quite similar to the one we obtain from our measurement. Ultimately, it is this enhancement factor, which is the crucial quantity for atmospheric calculations. As this paper has a clear experimental focus, we are confident that the similar results for the enhancement factor by Halonen et al., Lehtipalo et al., Kuerten et al. and our study do support the validity of our results.

*Also, as it is often repeated in the manuscript, the claim that the enhancement is solely due to Debye-type permanent dipole-induced dipole interaction should be rationalized somewhere. For a single sulfuric acid molecule and a spherical particle containing a large number of molecules, neglecting the Keesom interactions is a good approximation, however for the interaction of a single molecule and a small cluster this may not be the case. And in any case, the London dispersion forces are always present.*

We agree with Roope Halonen, that we did stress this type of interaction too strongly in the manuscript. In the revised version we refrain from calling the interaction Debye-type, but use the summarizing term van-der-Waals-forces, which includes London dispersion force, Keesom interaction and Debye interaction. We added a clarifying sentence on **line 127** in the revised manuscript.

**Response to Christopher Hogan, Short Comment posted on ACPD December 5, 2019**

Please refer to the answer posted during the public discussion phase of this article. Thanks to Christopher Hogan, for thoroughly checking our derivations. We corrected the misleading factor 2, applied also the approach of Fuchs and Sutugin (1965) and changed our analysis to mass diameter based. Please find the details about changes made to the manuscript above.

*Supplement of*
**Enhanced growth rate of atmospheric particles from sulphuric acid**

**Dominik Stolzenburg et al.**

*Correspondence to*: Paul M. Winkler (paul.winkler@univie.ac.at)

**S1 Experimental design**

The CERN CLOUD chamber (detailed description can be found in Duplissy et al. (2016)) is a 26.1 m$^3$ stainless steel aerosol chamber, which can be kept at a constant temperature within 0.1 K precision. The dry air supply for the chamber is provided by boil-off oxygen and boil-off nitrogen mixed at the atmospheric ratio of 79:21. This ensures extremely low contaminant levels, especially from organics and sulphuric acid. Water vapour, ozone and other trace gases, such as $SO_2$ can be precisely added at the pptv-level. For the experiments presented here, apart from $O_3$ (~120 ppb), $SO_2$ (5 ppb), $NH_3$ and water (38% or 60% relative humidity), the chamber was kept clean. The cleanliness from organic contaminants was validated by the measurement of chamber air with a PTR3 proton-transfer-reaction time of flight mass spectrometer (Breitenlechner et al., 2017) and a nitrate chemical ionization-atmospheric pressure interface-time of flight mass spectrometer (nitrate CI-APi-ToF) (Jokinen et al., 2012). The absence of any contamination from amines was confirmed by measurements with a water cluster-CI-APi-ToF (Pfeifer et al., 2019), which did not register dimethylamine mixing ratios above the detection limit of 0.1 pptv. Additionally, the CLOUD experiment offers the possibility to study new particle formation under different ionization levels. Two electrode-grids one at the top and one at the bottom of the chamber can be supplied with a $\pm 30$ kV high voltage. The generated electric field clears ions and charged particles from the chamber within seconds, in order to study growth by sulphuric acid under neutral conditions. If the clearing field is switched off, ionization from galactic cosmic rays naturally increase the ion concentration inside the chamber.

The experiments were initiated by homogeneous illumination of the chamber at constant $O_3$ and $SO_2$ levels. The UV light of four Hamamatsu UV lamps guided into the chamber with fibre optics induced the photo-dissociation of $O_3$ and production of OH· radicals. Thereby, $SO_2$ is oxidized, leading to the formation of sulphuric acid. Particle growth rates were measured with the appearance time method, which requires a growing particle population, which can be clearly identified (Lehtipalo et al., 2014; Stolzenburg et al., 2018). Therefore, after the new particle population reached 10 nm, a short 30 minute cleaning stage was performed in order to increase loss rates for particles and sulphuric acid. As shown for the typical run example in Fig. S1, this leads to clearly separated growing particle populations in subsequent experiments.

30 Sulphuric acid monomer concentrations were measured with a nitrate CI-API-TOF. The sulphuric acid concentration was determined as the mean of the sum of three different molecular ion signals ($HSO_4^-$, $HNO_3HSO_4^-$, $(H_2O)HSO_4^-$) during the time period where the growth rate was measured. The signal is normalized by the main nitrate reagent ions and corrected for sampling line losses. Calibration of the instrument's response to sulphuric acid (Kürten et al., 2012) was performed before and after the measurement campaign and yielded comparable results.

35 Compared to previous studies, also the measurement of gas-phase $NH_3$ significantly improved due to the deployment of a calibrated water cluster CI-APi-ToF (Pfeifer et al., 2019). The protonated water cluster reagent ions selectively ionize ammonia and amines at ambient pressure. The detection limits reach approximately 0.5 pptv for ammonia and 0.1 pptv for dimethylamine. The data measured by the water cluster CI-APi-ToF were cross-checked at high $NH_3$ concentrations against a commercial PICARRO $NH_3$ analyser.

40

Particle growth was monitored using a DMA-train (Stolzenburg et al., 2017) for the size-range of 1.8-8 nm and a TSI Model 3936 nano-SMPS for sizes larger than 5 nm. Both instruments use electrical mobility classification in order to infer a dry particle size-distribution. For the calculation of collision kinetics (see Section S2), mobility diameters are therefore first corrected to mass diameters (Larriba et al., 2011) and second a wet particle size is determined assuming a most probable water

45 content with the SAWNUC code (Ehrhart et al., 2016). Compared to the scanning particle-size-magnifier (see e.g. Lehtipalo et al., 2014), which was used as the main instrument for particle growth measurements in Lehtipalo et al. (2016), these instruments using direct mobility analysis have less systematic uncertainties on the actual size classification. The size-ranges of both studies are also not directly comparable. We show the measurements in the lower size-interval of the DMA-train (1.8-3.2 nm mobility diameter) together with the earlier results (size range 1.5-2.5 nm mobility diameter) in Fig. S2. The results

50 with ammonia added to the chamber, which presumably decreases the evaporation rates to close to zero, agree nicely with the collision enhanced kinetic limit of this study, when adjusting the size-range is adjusted. Particle growth rates are inferred with the appearance time method (Lehtipalo et al., 2014; Stolzenburg et al., 2018). Figure S1d demonstrates for a representative experiment how the signal in each size channel of the used mobility analysers is fitted by a sigmoidal shape curve in order to infer the 50 % appearance time, i.e. the time where 50 % of the maximum signal intensity during the run is reached first. The

55 appearance times are plotted versus the corresponding diameter and fitted with a linear function over the two size intervals 1.8-3.2 nm and 3.2-8 nm. Nanoparticle growth rates inferred via this method could be affected by a strong contribution of cluster coagulation (Kontkanen et al., 2016; Li and McMurry, 2018). However, the comparison with a modelling study in Fig. S3 indicates that this contribution is expected to be negligible. Moreover we also used the size- and time-resolving growth rate analysis method INSIDE (Pichelstorfer et al., 2018) which accounts for coagulation and wall losses to cross-check our results

60 (Fig. 4b).

**S2 Theory of growth caused by sulphuric acid condensation**

Our description of particle growth rates follows the approach of Nieminen et al., (2010), but with a modified kinetic coefficient (Chan and Mozurkewich, 2001). In addition, the effect of hydration of vapour and particles needs to be considered from an experimental perspective (see Section S1). Accordingly, the particle growth rate ($GR$) is defined as:

65
$$GR = \frac{\mathrm{d}d_{p,\mathrm{dry}}}{\mathrm{dt}} = \frac{\frac{\mathrm{d}V_{p,\mathrm{dry}}}{\mathrm{dt}}}{\frac{\mathrm{d}V_p}{\mathrm{d}d_{p,\mathrm{dry}}}} = \frac{k_{coll}(d_{v,\mathrm{hyd}}, d_{p,\mathrm{hyd}}) \cdot V_{v,\mathrm{dry}} \cdot C_v}{\frac{\mathrm{d}}{\mathrm{d}d_{p,\mathrm{dry}}}[\frac{\pi}{6}d_{p,\mathrm{dry}}^3]}, \tag{S1}$$

where $d_{p,\mathrm{dry}}$ is the growing dry particle mass diameter, $V_{p,\mathrm{dry}}$ and $V_{v,\mathrm{dry}}$ are the dry volume of particle and unhydrated vapour molecule, $C_v$ is the vapor monomer concentration and $k_{coll}(d_{v,\mathrm{hyd}}, d_{p,\mathrm{hyd}})$ is the kinetic collision frequency between particle and vapour which is calculated using the hydrated sizes of the monomer and growing particle. The assumptions on vapour and particle composition specified in Section 2.1 can be used to determine the hard-sphere kinetic collision frequency based on
70  vapour and particle density, as e.g. given in (Seinfeld and Pandis, 2016). The accommodation coefficient is assumed unity. We then additionally consider a collision enhancement of neutral vapour monomers and particles due to attractive van-der-Waals forces, which can either  where the collision frequency can be described according to Sceats (1989):

$$k_{coll}(d_{v,\mathrm{hyd}}, d_{p,\mathrm{hyd}}) = k_K \cdot \left( \sqrt{1 + \left(\frac{k_K}{2k_D}\right)^2} - \left(\frac{k_K}{2k_D}\right) \right), \tag{S2}$$

with $k_K = \frac{\pi}{4} \cdot (d_{v,\mathrm{hyd}} + d_{p,\mathrm{hyd}})^2 \cdot \left(\frac{8kT}{\pi}\right)^{1/2} \cdot \left(\frac{1}{m_{v,\mathrm{hyd}}} + \frac{1}{m_{p,\mathrm{hyd}}}\right)^{1/2} \cdot E(\infty)$ and $k_D = 2\pi \cdot (d_{v,\mathrm{hyd}} + d_{p,\mathrm{hyd}}) \cdot (D_v + D_p) \cdot E(0)$

75  as the kinetic collision rates for the free molecule and continuum regime, respectively. They depend on the diameters $d_{v/p}$ the masses $m_{v/p}$ and the diffusion coefficients $D_{v/p}$ of the colliding hydrated vapour molecules or particles, respectively. Eq. (S2) is designed such that it reaches the correct limits of the free molecular and diffusion regime comparable to the approach of (Fuchs and Sutugin, 1971). However, compared to Nieminen et al. (2010), it includes collision enhancement factors due to van-der-Waals forces, $E(\infty)$ and $E(0)$. These factors can be linked to the attractive potential of van-der-Waals forces
80  described by Hamaker (1937). For the continuum regime, this is done by solving the integral:

$$E(0) = \left[ \int_{(r_v+r_p)}^{\infty} \left(\frac{r_v+r_p}{x^2}\right) \exp\left(\frac{\phi(x)}{kT}\right) dx \right]^{-1} \tag{S3}$$

where $x$ is the relative distance between the centres of the two colliding entities and $\phi(x)$ is the van-der-Waals potential (Hamaker, 1937), which is expressed in terms of the vapour and particle radii $r_{v/p}$:

$$\frac{\phi(x)}{kT} = -\frac{1}{6}\frac{A}{kT}\left(\frac{2\,r_v r_p}{x^2-(r_v+r_p)^2} + \frac{2\,r_v r_p}{x^2-(r_v-r_p)^2} + \ln\left(\frac{x^2-(r_v+r_p)^2}{x^2-(r_v-r_p)^2}\right)\right)$$

85  Chan and Mozurkewich (2001) provide a fit to the numerical solution of the numerically evaluated integral from Sceats (1989):

$$E(0) = 1 + a_1 \cdot \ln(1 + A') + a_2 \cdot \ln^3(1 + A') \,, \tag{S4}$$

where $a_n$ and $b_m$ are the fit parameters and $A'$ is the reduced Hamaker constant, which relates to the Hamaker constant $A$ (Chan and Mozurkewich, 2001; Hamaker, 1937):

90 $$A' = \frac{A}{kT} \frac{4 d_v d_p}{(d_v + d_p)^2} \,, \tag{S5}$$

However, note that the measurements of this study are conducted completely in the free molecular regime, and hence the derivation of the continuum case will not affect our results. For the free molecular regime enhancement factor $E(\infty)$, the relation to the Hamaker constant is usually derived from ballistics arguments, where an overview is given in Ouyang et al. (2012). Chan and Mozurkewich (2001) also here used a fit to the solution from Sceats (1989):

95 $$E(\infty) = 1 + \frac{\sqrt{A'/3}}{1 + b_0 \sqrt{A'}} + b_1 \cdot \ln(1 + A') + b_2 \cdot \ln^3(1 + A') \,, \tag{S6}$$

Compared to the continuum regime, the relation between the Hamaker constant and the enhancement factor in the free molecular regime might depend on the chosen approach (Ouyang et al., 2012). In this study, we compare the results obtained by the fits of Chan and Mozurkewich (2001), i.e. the results of Sceats (1989), who used Brownian coagulation to describe the collisions, to the simple ballistics approach of Fuchs and Sutugin (1965). There, the minimum distance $x_{\min}$ along the

100 trajectory of two colliding particles with impact parameter $b$ is calculated from conservation of angular momentum und energy:

$$b = x_{\min} \sqrt{1 + \left(\frac{2|\phi(x_{min})|}{\mu v^2}\right)} \tag{S7}$$

where $\phi$ is the interaction potential, $\mu$ the reduced mass of the colliding entities and $v$ their relative speed. The critical impact parameter $b_{\mathrm{crit}}$ is obtained as the minimum value of $b$ for which the minimum distance still takes a real value larger than $(r_v + r_p)$. The enhancement factor is than related to the critical impact parameter $b_{\mathrm{crit}}$ :

105 $$E(\infty) = \frac{4 \, b_{\mathrm{crit}}^2}{(d_v + d_p)^2} \sqrt{\frac{3}{2}} \tag{S8}$$

Note, that this approach is oversimplified, as the initial velocity of the colliding entities is assumed to be fixed but should actually follow a distribution. Ouyang et al. (2012) however concluded that the difference in the derived Hamaker constant is almost negligible.

Altogether, we use the condensation equation Eq. (S1) and integrate it numerically over the size interval (dry mass diameter) used for the determination of the growth rate $[d_{\mathrm{init,dry}}, d_{\mathrm{final,dry}}]$:

110

$$GR\left(d_{\mathrm{init,dry}}, d_{\mathrm{final,dry}}\right) = \frac{\Delta d_{p,\mathrm{dry}}}{\Delta t} = \left(d_{\mathrm{final,dry}} - d_{\mathrm{init,dry}}\right) \Big/ \int_{d_{\mathrm{init,dry}}}^{d_{\mathrm{final,dry}}} \frac{\pi/2 \cdot d_p^2}{k_{coll}(d_{v,\mathrm{hyd}}, d_{p,\mathrm{hyd}}) \cdot V_{v,\mathrm{dry}} \cdot C_v} \, \mathrm{d}d_p \tag{S9}$$

where $k_{coll}$ can then be either the hard sphere collision rate, the enhanced collision rate Eq. (S2) linked to the Hamaker constant by Eq. (S6) (Chan and Mozurkewich approach) or by Eq. (S8) (Fuchs and Sutugin approach).

115   An analytical approximation of the growth rates such as that proposed by Nieminen et al. (2010) is not possible for the more complex collision enhanced collision kernel. Therefore, we parametrize the size- and sulphuric acid dependency of growth rates from 2-10 nm (mobility diameter) and 278-293 K by the empirical equation (maximum 8% discrepancy):

$$GR(\text{nm h}^{-1}) = \left[2.56 \cdot d_p(\text{nm})^{-1.41} + 1.03\right] \cdot [H_2SO_4(\text{cm}^{-3}) \cdot 10^{-7}] \tag{S10}$$

This parametrization does not include any possible humidity dependence, which was not significantly identified in our limited

120   dataset. In addition, note that this is only valid for dried aerosol particles using their mass diameter.

**S2.1 Assumed properties of the condensing clusters and systematic uncertainties**

Eq. (S9) includes several properties of the condensing vapour and the growing particles. As growth is measured for dried mobility classified particles, the added volume $V_v$ due to monomer condensation can be linked to its molecular mass and density $V_V = m_v/\rho_v$, with $m_v = 98$ amu and $\rho_v = 1849 \text{ g cm}^{-3}$ for sulphuric acid at 278.15 K. However, sulphuric acid

125   molecules and particles are usually hydrated at typical ambient relative humidity (Hanson and Eisele, 2000; Henschel et al., 2014; Kurtén et al., 2007). The macroscopic density of sulphuric acid can be derived from the mass fraction of the acid water solution (Myhre et al., 1998):

$$\rho_{v/p}(w, T)[\text{kg m}^{-3}] = \sum_{i=0}^{10} \sum_{j=0}^{4} \rho_{i,j} w^i \cdot (T[K] - 273.15)^j, \tag{S11}$$

where $w$ is the mass fraction of sulphuric acid in solution with $(1 - w)$ water. The coefficients $\rho_{ij}$ can be found in Myhre *et*

130   *al.* (Myhre et al., 1998). For the condensing monomers, the mass fraction is inferred from the assumed molecular composition of the nucleating clusters. At 298 K and 40-60% relative humidity Henschel et al. (2014) showed with quantum chemical calculations that the sulphuric acid monomer is on average hydrated by 1 water molecule, while the thermodynamic model E-AIM (Wexler et al., 2002) and the calculations of Kurtén et al. (2007) predict on average 2 attached water molecules for these conditions. This will result in a vapour molecular mass of $m_v = 134$ amu, which is used in the following. In the presence of

135   ammonia, nucleation of this ternary system proceeds with sulphuric acid in excess of ammonia (Hanson and Eisele, 2002): only one ammonia molecule is needed to stabilize a sulphuric acid dimer efficiently. If per two condensing sulphuric acid molecules, one ammonia molecule is bound to the particle, the mass per condensing cluster is increased by $m_{NH3}/2 = 8.5$ amu, which does not significantly influence the calculations and is therefore omitted in the following. The last property of the condensing vapour for the computation of Eq. (S9) is the diffusion coefficient of sulphuric acid. As also the diffusion

140   coefficient depends on the degree of hydration of the sulphuric acid molecule, we use relative humidity dependence of the diffusion coefficient for sulphuric acid as follows (Hanson and Eisele, 2000):

$$D_v(RH[\%]) = \frac{1}{p[\text{atm}]} \cdot \left(\frac{T[K]}{298}\right)^{1.75} \cdot \frac{p_0 D_0 + p_0 D_1 \cdot K_1 \cdot RH + p_0 D_2 \cdot K_1 K_2 \cdot (RH)^2}{1 + K_1 RH + K_1 K_2 (RH)^2}, \tag{S12}$$

which includes corrections for pressure and temperature different to the original experimental values. The fitting constants and the values for the diffusivity of the pure monomer, the singly hydrated monomer and the doubly hydrated monomer ($p_0 D_0$,

145   $p_0 D_1$, $p_0 D_2$, respectively) can be found in Hanson and Eisele (2000).

For the growing particles, it is necessary to infer the actual hydrated size, as particles are measured below 5 % relative humidity in the used mobility spectrometers. The increase in particle diameter due to water is usually described by the hygroscopic growth factor:

$$\text{gf} = \left( \frac{\rho_{\text{sol}}(w(5\%,T),T)}{w \cdot \rho_{\text{sol}}(w(RH,T),T)} \right)^{1/3}$$

150 where the density of the solution $\rho_{\text{sol}}$ can be obtained by Eq. (S11) if the mass fraction of sulphuric acid $w(RH,T)$ in the growing particles is known. Here, we use the SAWNUC code (Ehrhart et al., 2016; Lovejoy et al., 2004) in order to calculate the most probable water content of particles between 1.8-10 nm for all temperatures and relative humidities of this study. This is shown in Fig. S4a, where the mass fraction of sulphuric acid together with the growth factor for initial 5% relative humidity are plotted as function of size.

155 Finally, the diffusivity of the growing particles can be described using their measured mobility diameter and Stoke's law, including the dynamic viscosity of air $\eta$ and the Cunningham slip correction factor $C(d_p)$:

$$D_p = \frac{kT\,C(d_p)}{3\pi\eta d_p}, \tag{S13}$$

We list all used relevant constants and parameters for our derivation in Table S1.

**S2.2 Systematic uncertainty estimate**

160 Main systematic uncertainties in the estimate of the Hamaker constant might be related to the assumed properties of the condensing cluster and the growth factor of the measured dry particles. In order to explore the wide range of systematic uncertainties, we ran the fitting algorithm 1000 times, randomly assigning different parameters, summarized in Fig. S4b. With an equal probability we either use 1, 2 or 3 water molecules as assumed degree of hydration for the condensing cluster. For the growth factor of the measured particles, we estimate the error of the derived mass fractions from SAWNUC to be around 165 0.1, i.e. we assume a Gaussian distribution for an additive water content offset with sigma 0.1 and mean 0 for our uncertainty estimate. Independent of the assumed properties of the condensing cluster is the systematic uncertainty of the sulphuric acid concentrations, which is estimated by assigning a multiplicative offset to all sulphuric acid measurements, which follows a lognormal distribution with median 1 and shape parameter $\sigma_g = 0.5/3$, assuring that with 99.7% probability the resulting systematic sulphuric acid offset is within the interval of +50/-33 %, which represents the maximum systematic uncertainty in 170 the absolute sulphuric acid calibration (Kürten et al., 2012). Fig. S4b shows that the systematic uncertainty of the nitrate CI-APi-ToF measurement of $H_2SO_4$ is the largest source of uncertainty in our Hamaker constant estimate, but on a minor level also the assumed growth factor (i.e. sulphuric acid mass fraction) does result in an additional increase in the systematic uncertainty. Other systematic factors, i.e. the method in calculating the diffusivity of the vapour (e.g. the approach of Cox and Chapman (2001)), the method used for the density of vapour and particle (including ammonia according to Hyvärinen et al., 175 (2005)) or the assumption of an addition of 8.5 amu to the vapour mass in the presence of ammonia, were all found to have an insignificant effect on the result distribution and are thus not shown in Fig. S4b.

[revised manuscript text omitted]

355 **Figure S3: The cluster contribution to growth.** Sulphuric acid, which is contained in small molecular clusters and hence is unaccounted in the measured monomer concentration could potentially contribute to growth by cluster condensation. This effect of 'hidden' sulphuric acid (Lehtipalo et al., 2016) would impact our kinetic results. (a) shows the measured ratio of $H_2SO_4$ monomers and dimers during growth experiments at 278.15 K with varying ammonia concentrations. While absolute cluster concentrations measured with the nitrate CI-APi-ToF are subject to larger uncertainties for the $H_2SO_4$ dimer, the

360 constant monomer/dimer ratio with and without addition of ammonia is indicating a negligible clustering effect. (b) shows a comparison of the growth rate predictions at the kinetic limit of condensation for three different approaches at a diameter of 2.4 nm and 278.15 K versus sulphuric acid monomer concentration. The black dashed line shows the geometric limit of kinetic condensation based on a hard-sphere assumption (Nieminen et al., 2010). The red line and red area show the result of this study and its systematic uncertainty. The light green line and the dark green dashed line show the predictions of a model which

365 includes sulphuric acid/ammonia clustering and evaporation (Kürten, 2019), for 4 pptv and 2000 pptv ammonia (and assumed sulphuric acid density of $\rho=1.615$ g cm$^{-3}$), respectively. An additional clustering and therefore cluster contribution to growth at higher ammonia levels is not significant in the model.

[Figure]

370 **Figure S4: Systematic uncertainty estimate.** (a) The results from the SAWNUC code (Ehrhart et al., 2016; Lovejoy et al., 2004) for the mass fraction of sulphuric acid (red lines) and particle growth factor (blue lines) as function of mass diameter for the three different chamber conditions studied. Solid lines represent 293.15 K and 60 % relative humidity, dashed lines 278.15 K and 60 % relative humidity and dotted lines 278.15 K and 40 % relative humidity. (b) Results from a Monte-Carlo estimate of the systematic uncertainty of our Hamaker constant measurement. Every histogram represents the outcome of 1000

375 Hamaker constant fits to our dataset with randomly assigned systematic parameters. For every colour, other systematic uncertainties are additionally considered. The systematic uncertainty from the sulphuric acid measurement (the only considered uncertainty in the blue histogram) dominates the width of all the result distributions. The effects of water content of the condensing cluster (vapour mass fraction $w_v$, additionally considered in the yellow histogram) and particle sulphuric acid mass fraction ($w_p$, additionally considered in the green histogram) slightly broaden the distribution. Other tested systematic

380 uncertainties not shown in this panel, did not have any significant effect on the output distribution.

| Property | 293.15 K 60% RH | 278.15 K 60% RH | 278.15 K 38% RH | 293.15 K 5 % RH |
|---|---|---|---|---|
| $m_v$ vapour mass | 134 amu | 134 amu | 134 amu | - |
| $\rho_v$ vapour density | 1649 kg m$^{-3}$ | 1661 kg m$^{-3}$ | 1661 kg m$^{-3}$ | - |
| $D_v$ vapour diffusivity | $9.44 \cdot 10^{-6}$ m$^2$ s$^{-1}$ | $8.61 \cdot 10^{-6}$ m$^2$ s$^{-1}$ | $8.65 \cdot 10^{-6}$ m$^2$ s$^{-1}$ | - |
| $l_v$ vapour dipole moment | 2.84 Debye | 2.84 Debye | 2.84 Debye | - |
| $\alpha_v$ vapour polarizability | 6.2 Å$^3$ | 6.2 Å$^3$ | 6.2 Å$^3$ | - |
| $\epsilon_v$ vapour permittivity | 1 | 1 | 1 | - |
| $\rho_p(d_p)$ particle density | 1507 – 1346 kg m$^{-3}$ | 1489 – 1351 kg m$^{-3}$ | 1529 – 1418 kg m$^{-3}$ | 1668 – 1619 kg m$^{-3}$ |
| $gf(d_p)$ particle growth factor | 1.24 – 1.39 | 1.25 – 1.40 | 1.21 – 1.31 | 1 |
| $\epsilon_p$ particle permittivity | ~100 | ~100 | ~100 | - |

**Table S1: Used parameters.** Summary of the relevant parameters, for the calculation of the theoretical collision kernels and growth rate estimates for the three different chamber conditions investigated in this study. The fourth column represents the instrument condition, where particles are dried to below 5 % relative humidity. If a parameter depends on the particle size $d_p$, its variation between 1.5 and 7.7 nm (mass diameter) is given (DMA-train size range in mass diameter).

---

## Referee Report (RR1)

The paper has been improved and indeed more information on the experiment and the analysis can now be discerned.   There are some crucial issues in the main result that appear to be open to alternate analyses.  The change in the approach now taken by the authors is good but the use of dry GR introduces digressions and considerable confusion to this reviewer.  Below I calculate and report a 'wet' GR that the particles actually undergo in the experiment.  This alternate analysis suggests the enhancement in collisional rates due to van der Waals forces is less than their analysis suggests.  The authors' discussion of the issues raised in (1) and (2) will illuminate the best way forward.

(1)  The explicit inclusion of the particle's water content in the analysis is quite important for growth rates and the derived collision rate coefficients.  Yet there is only a modest change in the results!  Inspection of the previous eqn (S1) revealed that a factor of two was used 'to include collisions in both ways'.  This factor was not justified in the first version (thanks to Chris Hogan's close look) and now the authors have dropped it, claiming that another factor of two canceled it.  Can the authors explain the evolution of their thought in developing eqn (S1)?  Where did the 'collisions in both ways' idea come from?  It seems that previously they used the $k_{coll}$ from Niemenen et al.  It is not clear how the factor of two was canceled out in the last version and what is going on with the revised calculations.

(1b)  The assumption that the dry volume of the gaseous $H_2SO_4$ molecules can be assigned to the change in the dry volume of the particles has not been shown to be true.  Furthermore, $H_2SO_4$ hydration is a matter of wide variability according to the quantum chemistry studies; why choose Henschel over Temelso and how much of an effect does an alternate choice make?

(2)  The growth factor expression seems a bit odd and needs a reference.  It seems a more standard expression would have in the numerator the density $\rho$ time weight fraction $w$ of the appropriate RH in the instrument.  Looking at the data for 5 nm dry mass diameter, the numerator has a value of 1.68 to get the $gf$ in the figure.  This is the density of 76 wt % SA. Applying this $w$ to the numerator, $gf$ is 1.24 for a dry mass diameter of 5 nm.

(3)  Using the data in Figures S1(b,c, [SA]=2.1e7 cm$^{-3}$) and applying $gf$ of 1.2 and 1.28 for the 2.9 and 7.7 mass diameter, one gets an experimental GR of 3.2 nm/hr over the 3.2 to 8 nm dry mob. diameter range.  Compare this value to the GR calculated using the 'wet' GR from eqn. (7) of Verheggen and Mozurkewich (JGR, 2002), one gets a GR of 2.17 nm/hr for 48 wt. % SA (the SA-content for the midrange dry mass diameter of 5 nm): 1.95 nm/hr from the first term on the RHS, involving growth by SA uptake, and 0.22 nm/hr for the 2nd term, particle swelling due to the change in composition with size.  Incorporating the size of the condensing molecule into eqn (7) of Verh. and Moz. (2002), by applying a factor of $(1+d_v/d_p)^2$ to the first term and the total GR is 2.6 nm/hr, about 20 % less than the experimental GR.  This analysis has a hard-sphere GR that departs from experimental somewhat less than what appears to be in the paper.

(4) Composition data is incredibly important here and Fig. S4a is a welcome figure.  Yet the composition etc. inside the instrument is also needed (the numerator in the *gf* eqn).  Having said that, the Fig. S4a SAWNUC composition data has not really been put to any stringent tests: it was the nucleation rates - for cold conditions - that were verified in Ehrhart et al. 2016 (or have I missed something in that work?)  Variations in $\rho$ and *w* should be considered in the uncertainty analysis.  Also important in this is that ammonia was present which is not considered in SAWNUC, thus more uncertainty.

(5)  As to the data for growth between 1.8 and 3.2 nm mobility diameter.  The 1.8 nm data does not show fidelity to the assumed time dependence as there is a slow climb in the count rate over the hours (Fig. S1).  What is the physical reason for the assumed shape of the appearance curve (a systematic concern)?  Also, the composition of the 'dried' 1.8 nm particles would be most affected by charge, even the SAWNUC calculations allude to that (Ehrhart (2016) plot).  Ammonia might affect the smallest channels differently than it would the larger channels.  Aside from systematic biases, there is a significant random uncertainty in the appearance time for the small diameter channels, many (5 or 10?) minutes.  To better serve the reader, there needs to be two paragraphs added to the main text.  (i) An experimental paragraph on how the DMA's were deployed and the experimental conditions in them (e.g., 5 % RH is due to dry sheath gas diluting the incoming moist sample aerosol? Change in temperature upon sampling? Sampling arrangement?)  (ii) A paragraph describing the main assumptions in the appearance time method, e.g. answering the questions raised above.

(6) A paragraph in the Supplement describing the Inside method is needed. For example, how do values appear continuously and even at lower diameter then the measurements it is derived from?  Why was the TREND method not selected?  Pichelstorfer et al. notes that this method is good for capturing the leading edge of growing aerosol.  There is also a log normal method.  It is notable that there seems to be only partial agreement between these methods when comparing to experimental data.

(7) It is not clear how S9 (previously S6) was obtained nor what the authors mean in the different nomenclature between the terms on the LHS of eqns. S1 and S9.  Is one of them an average GR?

---

## Author Response (AR2)

**Response to the Anonymous Referee #2 on the revised version of "Enhanced growth rate of atmospheric particles from sulfuric acid"**

**Response to Anonymous Referee #2:**

*The paper has been improved and indeed more information on the experiment and the analysis can now be discerned. There are some crucial issues in the main result that appear to be open to alternate analyses. The change in the approach now taken by the authors is good but the use of dry GR introduces digressions and considerable confusion to this reviewer. Below I calculate and report a 'wet' GR that the particles actually undergo in the experiment. This alternate analysis suggests the enhancement in collisional rates due to van der Waals forces is less than their analysis suggests. The authors' discussion of the issues raised in (1) and (2) will illuminate the best way forward.*

We are thankful to the reviewer for the careful analysis of our updated calculations. We feel that an even more thorough discussion of the effects of water will improve this manuscript. The main change in the revised manuscript is to compare three approaches describing hygroscopicity and to revise Figure 4b. As the figure shows, these three very different approaches to treating the effects of water have little quantitative and negligible qualitative effect on our conclusions. We therefore argue that our conclusions are robust. We also agree with the reviewer that several aspects still need to be clarified and thus we re-organized the manuscript by including large parts from the Supplement into a Methods section in order to ensure all key information is present in the main text. We also updated all Figures to have a more publication-ready format and hope this improves the presentation quality.

*(1) The explicit inclusion of the particle's water content in the analysis is quite important for growth rates and the derived collision rate coefficients. Yet there is only a modest change in the results! Inspection of the previous eqn (S1) revealed that a factor of two was used to "include collisions in both ways". This factor was not justified in the first version (thanks to Chris Hogan's close look) and now the authors have dropped it, claiming that another factor of two canceled it. Can the authors explain the evolution of their thought in developing eqn (S1)? Where did the"collisions in both ways" idea come from? It seems that previously they used the kcoll from Niemenen et al. It is not clear how the factor of two was canceled out in the last version and what is going on with the revised calculations.*

In the answer to Chris Hogan's comment, we showed the origin of the error (taking the kernels from Chan & Mozurkewich 2001), and the reason why the factor did cancel already in the first place (a factor of 2 used in front of the collision kernel). This is well summarized in the public response to Chris Hogan. We corrected the equations, deleted the sentence of the "collision in both ways" and updated the calculations. We also modified the entire theory section of the growth rate derivation to give it a more logical structure in the revised version. Thus we are confident that the reader of the revised manuscript will understand how the calculations have been performed.

*(1b) The assumption that the dry volume of the gaseous H2SO4 molecules can be assigned to the change in the dry volume of the particles has not been shown to be true. Furthermore, H2SO4 hydration is a matter of wide variability according to the quantum chemistry studies; why choose Henschel over Temelso and how much of an effect does an alternate choice make?*

*(2) The growth factor expression seems a bit odd and needs a reference. It seems a more standard expression would have in the numerator the density times weight fraction w of the appropriate RH in the instrument. Looking at the data for 5 nm dry mass diameter, the numerator has a value of 1.68 to get the gf in the figure. This is the density of 76 wt % SA. Applying this w to the numerator, gf is 1.24 for a dry mass diameter of 5 nm.*

*(3) Using the data in Figures S1(b,c, [SA]=2.1e7 cm-3) and applying gf of 1.2 and 1.28 for the 2.9 and 7.7 mass diameter, one gets an experimental GR of 3.2 nm/hr over the 3.2 to 8 nm dry mob. diameter range. Compare this value to the GR calculated using the 'wet' GR from eqn. (7) of Verheggen and Mozurkewich (JGR, 2002), one gets a GR of 2.17 nm/hr for 48 wt. % SA (the SA-content for the midrange dry mass diameter of 5 nm): 1.95 nm/hr from the first term on the RHS, involving growth by SA uptake, and 0.22 nm/hr for the 2nd term, particle swelling due to the change in composition with size. Incorporating the size of the condensing molecule into eqn (7) of Verh. and Moz. (2002), by applying a factor of (1+dv/dp)2 to the first term and the total GR is 2.6 nm/hr, about 20 % less than the experimental GR. This analysis has a hardsphere GR that departs from experimental somewhat less than what appears to be in the paper.*

These three points help to clarify the effect of water on our measured growth rates. The reviewer is correct that the assumptions of dry particles at 5 % RH is questionable and that the numerator in Eq. (13) of the revised manuscript needs to be defined differently (the numerator needs to include the mass fraction of the measurement, see e.g. Biskos et al., 2009, J. Aerosol Sci.). We also appreciate the reviewer pointing us towards the paper of Verheggen and Mozurkewich (2002), which offers a different view on the problem. As also pointed out by the reviewer, composition data are important if we follow this approach. This is true for both chamber and measurement conditions. We hence decided to compare three approaches in the revised manuscript and discuss their agreement with the measurement data in an entirely new section in the revised manuscript.

A first naïve approach which follows the first version of the manuscript, i.e. taking an average hydration of the monomer and assuming this to be the same for the growing particle, which is unaltered during the measurement. Here it is important to note that the results from Temelso 2012, Henschel 2014, Kurten 2007 and Wexler and Clegg 2002 actually agree quite well, all predicting an average hydration of either 1 or 2 $H_2O$ molecules at around 50% RH. Overall, for the conditions of our measurements, we do not find the variety of hydration results in the recent literature, contrary to the assertion of the reviewer. Most probable hydration is either 1 or 2 water molecules per sulfuric acid under our experimental conditions and this is covered in our uncertainty estimate already.

Second, we adjusted the naïve measurement approach, by acknowledging the fact that the actual particle size is measured dry. For this we assume a hygroscopic growth factor of 1.25, which is the average value of the results of Biskos et al. (2009), J. Aerosol Sci., for sub-10 nm particles at 40-60% RH. All HTDMA studies, which have served as a basis for composition data, base their measured growth factors also from a dry size at or around 5 % RH, i.e. close to the conditions of the DMA-train, which allows us to choose such a value from literature. The reviewer is correct that the mass addition per collision to a particle at 5% RH might still be influenced by water, but composition data for 5% RH is basically not available from measurements, but only from models. Here, e.g. MABNAG predicts almost no hydration at 5 % RH for particles larger than 3 nm, so the assumption of a dry measurement does not seem to be a huge oversimplification.

Third, we followed the approach of Verheggen & Mozurkewich (2002) and used both SAWNUC and MABNAG to both predict hydration of particles in the chamber and during the measurement. In this approach the effect of water uptake and sulfuric acid driven growth are separated (thus called separation approach in our revised manuscript). All three approaches yield similar results for the collision enhancement (and the Hamaker constant), if we use the MABNAG composition data. We show a comparison of the approaches with the measured growth rates in Fig. 4b of the revised manuscript. We also show the results when using SAWNUC for the separation approach, which do not agree well with our measurements. While SAWNUC certainly estimates the hydration of the smaller clusters better than MABNAG (which was already identified in Yli-Juuti et al., 2013, ACP), SAWNUC might predict a too low sulphuric acid mass fraction at larger sizes. This could be mainly due to the effect that even at the low ammonia levels of the measurements, neutralization occurs to some extend as predicted by MABNAG. Hence, we come to the conclusion that the constant growth factor assumed in the dry measurement approach might give a good estimate at small and larger sizes and that additional swelling by water might be a minor effect.

Altogether, all three approaches yield a Hamaker constant which is well within our overall systematic uncertainty estimate (note that all of them are within the uncertainty estimate of the first version of this paper). We therefore come to the final conclusion that, independent of the approach, the Hamaker constant can be well estimated by our final given value and uncertainty range of $A = 5.2^{+9.7}_{-3.4}(syst.) \cdot 10^{-20}$J.

*(4) Composition data is incredibly important here and Fig. S4a is a welcome figure. Yet the composition etc. inside the instrument is also needed (the numerator in the gf eqn). Having said that, the Fig. S4a SAWNUC composition data has not really been put to any stringent tests: it was the nucleation rates - for cold conditions - that were verified in Ehrhart et al. 2016 (or have I missed something in that work?) Variations in rho and w should be considered in the uncertainty analysis. Also important in this is that ammonia was present which is not considered in SAWNUC, thus more uncertainty.*

The reviewer is correct that the assumed composition data from SAWNUC might be one of the weaknesses in this approach and that they were verified with data from 278 K downwards. Especially SAWNUC does not include ammonia, which could cause some neutralization of the small particles even at concentrations as low as 3 pptv. We therefore also present the predicted the particle hydration by calculations with MABNAG in the revised manuscript.

Variations in w (and hence in rho) were already considered in the last version of the manuscript. Nevertheless, they are again included and discussed in our updated Figure S5 in the Supplement.

*(5) As to the data for growth between 1.8 and 3.2 nm mobility diameter. The 1.8 nm data does not show fidelity to the assumed time dependence as there is a slow climb in the count rate over the hours (Fig. S1). What is the physical reason for the assumed shape of the appearance curve (a systematic concern)? Also, the composition of the 'dried' 1.8 nm particles would be most affected by charge, even the SAWNUC calculations allude to that (Ehrhart (2016) plot). Ammonia might affect the smallest channels differently than it would the larger channels. Aside from systematic biases, there is a significant random uncertainty in the appearance time for the small diameter channels, many (5 or 10?) minutes. To better serve the reader, there needs to be two paragraphs added to the main text. (i) An experimental paragraph on how the DMA's were deployed and the experimental conditions in them (e.g., 5 % RH is due to dry sheath gas diluting the incoming moist sample aerosol? Change in temperature upon sampling? Sampling arrangement?) (ii) A paragraph describing the main assumptions in the appearance time method, e.g. answering the questions raised above.*

We respectfully disagree with the reviewer. First, we do not see any significant climb in count rate over the hours in Figure S1. There are some small fluctuations which might come from fluctuations in the nucleation rate (see also the slowly increasing trend of ammonia over the same time-span) and these are also visible in the other size channels. The appearance time fit however is chosen to cover a similar time-range for all channels, making the linear fit robust. Hence, we are highly confident that the agreement between the appearance time fit and the measured data is sufficient to provide a good growth rate measurement. The appearance time method is well-documented in literature and hence we

do not see the need the elaborate in detail on this but refer to a recent published article in Nature Protocols (Dada et al., Nat. Protoc., 2020) in the Methods Section. Second, in our opinion, the effect of charge on the hydration at 1.8 nm is not significant in the Ehrhart et al. (2016) plot (the effect of charge vanishes above 1.5 nm). Moreover, the reported growth rates are integrated growth rates from 1.8-3.2 nm and a close-to-insignificant effect at 1.8 nm would hardly bias the entire growth rate result, which we clarified by stating that the measured growth rates are integrated growth rate from 1.8-3.2 nm in the revised Methods Section. However, the reviewer is correct that the assumed time dependence in the appearance time method fits has no physical basis, which we added in the Methods Section. Certainly, the appearance time method does suffer from systematic uncertainties, and hence we included the calculations using INSIDE. Here the approach is entirely different and the excellent agreement between both methods gives us the high confidence on our dataset. We also agree with the reviewer that we should specify the measurement methods in more detail with respect to the open questions about hydration and we have updated the Methods Section accordingly.

*(6) A paragraph in the Supplement describing the Inside method is needed. For example, how do values appear continuously and even at lower diameter then the measurements it is derived from? Why was the TREND method not selected? Pichelstorfer et al. notes that this method is good for capturing the leading edge of growing aerosol. There is also a log normal method. It is notable that there seems to be only partial agreement between these methods when comparing to experimental data.*

As the reviewer suggested, we extended our description on the INSIDE method in the Supplement. TREND and INSIDE agree nicely in the size range where both methods yield results. However, INSIDE is evaluated at the same diameters over a larger size range compared to TREND, we used INSIDE to average over an entire run. Note, that also no INSIDE results are reported below 1.8 nm (mobility diameter), so we do not share reviewer's concerns on this. For the problems of the applicability of the lognormal method to chamber data we refer the reviewer to Dada et al. (Nat. Protoc. 2020, p.11): "(…), including the log-normal distribution function method (which is not covered in this protocol because it is often unsuitable for chamber experiments, being that there are no distinct particle modes)".

*(7) It is not clear how S9 (previously S6) was obtained nor what the authors mean in the different nomenclature between the terms on the LHS of eqns. S1 and S9. Is one of them an average GR?*

This has been clarified in the revised theoretical growth rate section of the revised manuscript.

[revised manuscript text omitted]